# DiffKendall: A Novel Approach for Few-Shot Learning with Differentiable Kendall's Rank Correlation

**Kaipeng Zheng**[1]    **Huishuai Zhang**[2]    **Weiran Huang**[1,†]

[1] Qing Yuan Research Institute, SEIEE, Shanghai Jiao Tong University
[2] Microsoft Research Asia

`kaipengm2@gmail.com, huzhang@microsoft.com, weiran.huang@outlook.com`

## Abstract

Few-shot learning aims to adapt models trained on the base dataset to novel tasks where the categories were not seen by the model before. This often leads to a relatively concentrated distribution of feature values across channels on novel classes, posing challenges in determining channel importance for novel tasks. Standard few-shot learning methods employ geometric similarity metrics such as cosine similarity and negative Euclidean distance to gauge the semantic relatedness between two features. However, features with high geometric similarities may carry distinct semantics, especially in the context of few-shot learning. In this paper, we demonstrate that the importance ranking of feature channels is a more reliable indicator for few-shot learning than geometric similarity metrics. We observe that replacing the geometric similarity metric with Kendall's rank correlation only during inference is able to improve the performance of few-shot learning across a wide range of methods and datasets with different domains. Furthermore, we propose a carefully designed differentiable loss for meta-training to address the non-differentiability issue of Kendall's rank correlation. By replacing geometric similarity with differentiable Kendall's rank correlation, our method can integrate with numerous existing few-shot approaches and is ready for integrating with future state-of-the-art methods that rely on geometric similarity metrics. Extensive experiments validate the efficacy of the rank-correlation-based approach, showcasing a significant improvement in few-shot learning.

## 1   Introduction

Deep learning has achieved remarkable success in various domains. However, obtaining an adequate amount of labeled data is essential for attaining good performance. In many real-world scenarios, obtaining sufficient labeled data can be exceedingly challenging and laborious. This makes it a major bottleneck in applying deep learning models in real-world applications. In contrast, humans are able to quickly adapt to novel tasks by leveraging prior knowledge and experience, with only a very small number of samples. As a result, few-shot learning has received increasing attention recently.

The goal of few-shot learning is to adapt models trained on the base dataset to novel tasks where only a few labeled data are available. Previous works mainly focus on classification tasks and have been extensively devoted to metric-learning-based methods [1, 2, 3, 4]. In particular, they learn enhanced embeddings of novel samples by sampling tasks with a similar structure to the novel task from a sufficiently labeled base dataset for training. Geometric similarity metrics, such as negative Euclidean distance [1] and cosine similarity [4, 5, 6, 7], are commonly utilized to determine the semantic

---

[†]Correspondence to Weiran Huang.

37th Conference on Neural Information Processing Systems (NeurIPS 2023).

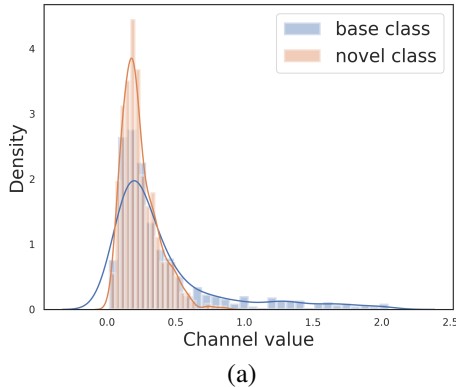
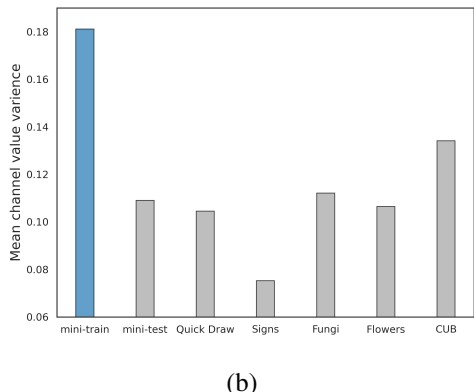

| (a) | (b) |

Figure 1: (a) Distribution of the feature channel values for a base class and a novel class on mini-ImageNet. The novel class is previously unknown for the model, where a clear difference can be observed from the base class. (b) Comparison of the Mean Variance of Feature Channel Values between the base dataset (mini-train) and various distinct new datasets. It is observed that the variance of feature channel values on the new datasets is consistently much lower than that on the base dataset.

similarities between feature embeddings. Recent studies [8, 9, 10] also show that simply employing cosine similarity instead of inner product in linear classifier also yields competitive performance of novel tasks, when the model is pre-trained on the base dataset.

Since the categories of the base class and novel class are disjoint in few-shot learning, there exists a significant gap between training and inference for the model. We found that compared to base classes, when the feature extractor faces a novel class that is unseen before, the feature channel values become more concentrated, i.e., for a novel class, most non-core features' channels have small and closely clustered values in the range [0.1, 0.3] (see Figure 1(a)). This phenomenon occurs because the model is trained on the base data, and consequently exhibits reduced variation of feature values when dealing with novel data. Empirically, we additionally compare the variance of feature channel values between the base dataset and various novel datasets (see Figure 1(b)). The results reveal a significantly smaller variance in feature channel values in the novel datasets compared to the base dataset. A smaller variance means values are closer to each other, demonstrating that this is a universally valid conclusion. This situation creates a challenge in employing geometric similarity to accurately distinguish the importance among non-core feature channels. To provide a concrete example, consider distinguishing between dogs and wolves. While they share nearly identical core visual features, minor features play a vital role in differentiating them. Suppose the core feature, two minor features are represented by channels 1, 2, and 3, respectively, in the feature vector. A dog prototype may have feature (1, 0.28, 0.2), and a wolf prototype may have feature (1, 0.25, 0.28). Now, for a test image with feature (0.8, 0.27, 0.22), it appears more dog-like, as the 2nd feature is more prominent than the 3rd. However, cosine distance struggles to distinguish them clearly, misleadingly placing this test image closer to the wolf prototype (distance=0.0031) rather than the dog prototype (distance=0.0034). Contrastingly, the importance ranking of feature channels is able to distinguish dogs and wolves. The test image shares the same channel ranking (1, 2, 3) as the dog prototype, whereas the wolf prototype's channel ranking is (1, 3, 2).

Motivated by the above observations, we aim to boost the few-shot learning based on the importance ranking of feature channels in this paper. Specifically, we propose a simple and effective method, which replaces the commonly-used geometric similarity metric (e.g., cosine similarity) with Kendall's rank correlation to determine how closely two feature embeddings are semantically related. We demonstrate that using Kendall's rank correlation at test time can lead to performance improvements on a wide range of datasets with different domains. Furthermore, we investigate the potential benefits of employing Kendall's rank correlation in episodic training. One main challenge is that the calculation of channel importance ranking is non-differentiable, which prevents us from directly optimizing Kendall's rank correlation for training. To address this issue, we propose a smooth approximation of Kendall's rank correlation and hence make it differentiable. We verify that using

the proposed differentiable Kendall's rank correlation at the meta-training stage instead of geometric similarity achieves further performance improvements.

In summary, our contributions to this paper are threefold as follows: 1) We reveal an intrinsic property of novel sample features in few-shot learning, whereby the vast majority of feature values are closely distributed across channels, leading to difficulty in distinguishing their importance; 2) We demonstrate that the importance ranking of feature channels can be used as a better indicator of semantic correlation in few-shot learning. By replacing the geometric similarity metric with Kendall's rank correlation at test time, significant improvements can be observed in multiple few-shot learning methods on a wide range of popular benchmarks with different domains; 3) We propose a differentiable loss function by approximating Kendall's rank correlation with a smooth version. This enables Kendall's rank correlation to be directly optimized. We verify that further improvements can be achieved by using the proposed differentiable Kendall's rank correlation at the meta-training stage instead of geometric similarity.

## 2 Related Works

**Meta-Learning-Based Few-Shot Learning.** Previous research on few-shot learning has been extensively devoted to meta-learning-based methods. They can be further divided into optimization-based methods [11, 12, 13] and metric learning-based methods [1, 2, 3, 4, 14, 15, 16], with metric learning-based methods accounting for the majority of them. ProtoNets [1] proposes to compute the Euclidean distance between the embedding of query samples and the prototypes on the support set for nearest-neighbor classification. Meta-baseline [4] uses cosine similarity instead and proposes a two-stage training paradigm consisting of pre-training and episodic training, achieving competitive performance compared to state-of-the-art methods. Recent research [3, 17] has begun to focus on aligning support and query samples by exploiting the local information on feature maps. Li et al. [17] represent support samples with a set of local features and use a $k$-nearest-neighbor classifier to classify query samples. CAN [3] uses cross-attention to compute the similarity score between the feature maps of support samples and query samples. ConstellationNet [6] proposes to learn enhanced local features by constellation models and then aggregate the features through self-attention blocks. Several studies [7, 18] investigate methods for calibrating novel sample features in few-shot learning. Xue and Wang [18] learn a transformation that moves the embedding of novel samples towards the class prototype by episodic training. Zhang et al. [7] propose to use additional attribute annotations that are generalizable across classes to complete the class prototypes. However, this requires additional labeling costs and the common properties will not exist when the categories of base class data and novel class data are significantly different. Unlike previous studies, we explore determining semantic similarities between novel sample features with the correlation of channel importance ranking in few-shot learning, which has never been studied before.

**Transfer-Learning-Based Few-Shot Learning.** Transfer-learning-based few-shot learning methods have recently received increasingly widespread attention. Prior research [8, 9, 10] has verified that competitive performance in few-shot learning can be achieved by pre-training models on the base dataset with cross-entropy loss and using cosine similarity for classification on novel tasks, without relying on elaborate meta-learning frameworks. Recent studies [19, 20, 21] also focus on introducing data from the pre-training phase to assist in test-time fine-tuning. For instance, Afrasiyabi et al. [19] propose to introduce samples from the base dataset in fine-tuning that are similar to the novel samples in the feature space. POODLE [21], on the other hand, uses the samples in the base dataset as negative samples and proposes to pull the embedding of novel samples away from them during fine-tuning.

## 3 Problem Definition

Typically, a few-shot task $\mathcal{T}$ consists of a support set $\mathcal{S}$ and a query set $\mathcal{Q}$. The objective of the few-shot task is to accurately predict the category for each query sample $x_i \in \mathcal{Q}$ based on the support set $\mathcal{S}$. The support set $\mathcal{S}$ is commonly organized as $N$-way $K$-shot, which means that there are a total of $N$ categories of samples included in this task, with each class containing $K$ annotated samples. In few-shot scenarios, $K$ is usually a very small number, indicating that the number of samples available for each category is extremely small.

It is infeasible to train a feature extractor $f_\theta$ on a few-shot task directly from scratch. Thus, the feature extractor $f_\theta$ is usually trained on a base dataset $\mathcal{D}_{\text{base}}$ with sufficient annotation to learn a prior. The training process typically involves pre-training and episodic training. After that, few-shot tasks are sampled on the novel dataset $\mathcal{D}_{\text{novel}}$ for performance evaluation. The categories of the samples in $\mathcal{D}_{\text{base}}$ are entirely distinct from those in $\mathcal{D}_{\text{novel}}$. The trained feature extractor $f_\theta$ is then used to obtain the embedding of the samples for novel tasks, followed by calculating the similarity between the embedding of support samples and query samples for a nearest-neighbor classification, namely,

$$P(y = k \mid \boldsymbol{x}) = \frac{\exp(\text{sim}(f_\theta(\boldsymbol{x}), \boldsymbol{c}_k) \cdot t)}{\sum_{j=1}^{N} \exp(\text{sim}(f_\theta(\boldsymbol{x}), \boldsymbol{c}_j) \cdot t)}, \tag{1}$$

where $\boldsymbol{x} \in \mathcal{Q}$ denotes the query sample, $\boldsymbol{c}_k$ represents the class prototype, which is usually represented by the mean feature of the support set samples in each category, $\text{sim}(\cdot)$ denotes a similarity measure, $N$ denotes the total number of classes included in the few-shot task, and $t$ is used to perform a scaling transformation on the similarities.

In supervised learning, since there is no distribution gap between the training and test data, we can obtain high-quality embedding of test data by training the model with a large number of in-domain samples. However, in few-shot scenarios, the categories in the training data and the test data have no overlap. As there is no direct optimization towards the novel category, the features of novel samples in few-shot learning can be distinct from those learned through conventional supervised learning [7, 18, 22].

Geometric similarities are commonly used to determine semantic similarities between features in few-shot learning, among which cosine similarity is widely employed in recent studies [4, 6, 8]. Given two vectors with the same dimension $\boldsymbol{x} = (x_1, ..., x_n)$ and $\boldsymbol{y} = (y_1, ..., y_n)$, denoting the features of novel samples, cosine similarity is calculated as $\frac{1}{\|\boldsymbol{x}\| \cdot \|\boldsymbol{y}\|} \sum_{i=1}^{n} x_i y_i$. It can be seen that the importance of each channel is directly correlated with the numerical feature value $x_i$ and $y_i$. However, as shown in Figure 1(a), the feature distribution of the novel class samples are largely different from that of the base class samples. The values of the feature channels are highly clustered around very small magnitudes, making it difficult for cosine similarity to differentiate their importance in classification. Consequently, the classification performance will be dominated by those very few channels with large magnitude values, while the small-valued channels, which occupy the majority of the features, will be underutilized. Although the embedding has already been projected onto the unit sphere in cosine similarity to reduce this effect, we verify that the role of the small-valued channels in classification is still largely underestimated.

## 4   Warm-Up: Using Kendall's Rank Correlation During Inference

In this paper, for the first time, we explore the utilization of channel importance ranking in few-shot learning. Converting numerical differences into ranking differences enables effective discrimination between small-valued channels that exhibit similar values, and reduces the large numerical differences between large-valued and small-valued channels. To achieve this, we start with investigating the use of Kendall's rank correlation in few-shot learning. Kendall's rank correlation gauges the semantic similarity between features by assessing how consistently channels are ranked, which aligns precisely with our motivation. In this section, we demonstrate that leveraging Kendall's rank correlation simply during the inference stage can lead to a significant performance improvement.

### 4.1   Kendall's Rank Correlation

Given two $n$-dimensional feature vectors $\boldsymbol{x} = (x_1, ..., x_n)$, $\boldsymbol{y} = (y_1, ..., y_n)$, Kendall's rank correlation is determined by measuring the consistency of pairwise rankings for every channel pair $(x_i, x_j)$ and $(y_i, y_j)$. This coefficient can be defined as the disparity between the number of channel pairs $(x_i, x_j)$ and $(y_i, y_j)$ that exhibit concordant ordering versus discordant ordering, namely,

$$\tau(\boldsymbol{x}, \boldsymbol{y}) = \frac{N_{\text{con}} - N_{\text{dis}}}{N_{\text{total}}}, \tag{2}$$

where $N_{\text{con}}$ represents the count of channel pairs with consistent importance ranking, i.e., either $(x_i > x_j) \wedge (y_i > y_j)$ or $(x_i < x_j) \wedge (y_i < y_j)$, $N_{\text{dis}}$ reflects the count of channel pairs with inconsistent ordering represented by either $(x_i > x_j) \wedge (y_i < y_j)$ or $(x_i < x_j) \wedge (y_i > y_j)$. $N_{\text{total}}$ represents the total number of channel pairs.

Table 1: Performance improvements by using Kendall's rank correlation at test time. The training set of mini-ImageNet is used as the base dataset and the average accuracy (%) of randomly sampled 5-way 1-shot tasks on test sets with different domains is reported.

| Method | Backbone | mini-test | CUB | Traffic Signs | VGG Flowers | Quick Draw | Fungi |
|---|---|---|---|---|---|---|---|
| CE + cosine | Conv-4 | 48.57 | 36.97 | 38.89 | 59.92 | 45.75 | 35.99 |
| CE + CIM | Conv-4 | 48.94 | 37.41 | **39.35** | 59.79 | 45.56 | 35.89 |
| CE + Kendall | Conv-4 | **51.50** | **39.01** | 39.04 | **61.55** | **46.00** | **36.73** |
| CE + cosine | ResNet-12 | 62.2 | 45.12 | 55.54 | 69.41 | 53.73 | 40.68 |
| CE + CIM | ResNet-12 | 60.4 | 45.20 | 56.64 | 69.61 | 55.14 | 40.34 |
| CE + Kendall | ResNet-12 | **63.3** | **47.07** | **60.84** | **71.38** | **55.99** | **41.68** |
| Meta-B + cosine | ResNet-12 | 62.84 | 45.38 | 54.88 | 69.14 | 53.27 | 40.57 |
| Meta-B + CIM | ResNet-12 | 61.60 | 45.24 | 55.31 | 68.87 | 54.08 | 40.41 |
| Meta-B + Kendall | ResNet-12 | **63.36** | **47.15** | **59.70** | **70.57** | **55.78** | **41.70** |
| CE + cosine | ResNet-18 | **62.92** | 43.7 | 47.17 | 62.35 | 52.33 | 38.89 |
| CE + CIM | ResNet-18 | 61.91 | 43.75 | 47.23 | 61.89 | 52.21 | 39.07 |
| CE + Kendall | ResNet-18 | 62.83 | **45.54** | **54.32** | **67.08** | **54.15** | **39.64** |
| CE + cosine | WRN-28-10 | 60.08 | 43.64 | 47.01 | 66.03 | 47.99 | 39.27 |
| CE + CIM | WRN-28-10 | 59.34 | 43.43 | 46.30 | 64.42 | 48.37 | 39.42 |
| CE + Kendall | WRN-28-10 | **61.68** | **45.80** | **51.39** | **69.72** | **53.52** | **41.52** |
| S2M2 + cosine | WRN-28-10 | **64.52** | 47.44 | 52.30 | 68.93 | 51.41 | 41.76 |
| S2M2 + CIM | WRN-28-10 | 63.60 | 47.59 | 53.84 | 70.91 | 53.89 | 42.54 |
| S2M2 + Kendall | WRN-28-10 | 63.97 | **47.74** | **57.88** | **71.48** | **54.63** | **43.49** |
| Avg Improvements (Kendall vs. cosine) | | 0.92 ↑ | 1.69 ↑ | 4.56 ↑ | 2.67 ↑ | 2.60 ↑ | 1.27 ↑ |
| Avg Improvements (Kendall vs. CIM) | | 1.81 ↑ | 1.63 ↑ | 4.13 ↑ | 2.88 ↑ | 1.80 ↑ | 1.18 ↑ |

## 4.2 Performace Improvements by Using Kendall's Rank Correlation at Test Time

We conduct comprehensive experiments and verify that directly using Kendall's rank correlation at test time can significantly improve performance in few-shot learning.

Specifically, recent studies [8, 9, 10] have confirmed that pre-training the model on the base dataset with cross-entropy loss, and utilizing cosine similarity for classification on novel tasks, can yield competitive performance. This approach has proven to outperform a number of meta-learning-based methods. Therefore, our initial comparison involves assessing the performance of cosine similarity and Kendall's rank correlation when the model is pre-trained with cross-entropy loss (CE) on the base dataset. The evaluation is conducted on different backbone networks that are commonly used in previous studies, including Conv-4, ResNet-12, ResNet-18, and WRN-28-10. In addition, the comparison is also made based on Meta-Baseline (Meta-B) [4], a representative meta-learning based approach in few-shot learning, and an advanced method S2M2 [5]. We keep all other settings unchanged and replace the originally utilized cosine similarity with Kendall's rank correlation solely during the testing phase. Furthermore, we expand our comparison to include a recently proposed method (CIM) [22], which suggests a test-time transformation for feature scaling in few-shot learning. The train set of mini-ImageNet [23] is used as the base dataset to train the model. Since the novel task confronted by the model can be arbitrary in real-world applications, we employ a wide range of datasets for testing that exhibit significant differences in both category and domain, including the test set of mini-ImageNet (mini-test), CUB [24], Traffic Signs [25], VGG Flowers [26], Quick Draw [27] and Fungi [28]. In the testing phase, we randomly sample 2000 few-shot tasks from the test dataset in the form of 5-way 1-shot and report the average accuracy on all tasks for performance evaluation. The results are shown in Table 1. It can be seen that simply using Kendall's rank correlation at test time in few-shot learning achieves significant improvements compared with cosine similarity. It also vastly outperforms the latest test-time feature transformation method CIM.

Moreover, we also demonstrate the effectiveness of Kendall's rank correlation in few-shot learning through another closer observation of performance. Specifically, we pre-train the model on the base dataset from scratch using standard cross-entropy loss and make a comparison between the performance of Kendall's rank correlation and cosine similarity on few-shot tasks after each training epoch. The results are shown in Figure 2. It can be seen that almost throughout the entire training

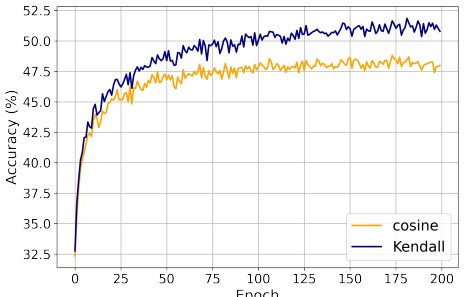 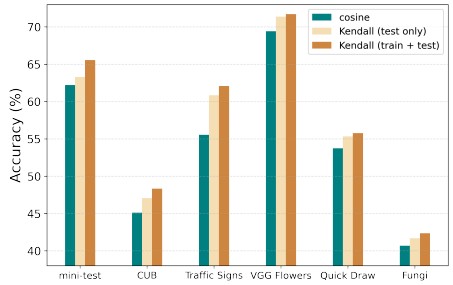

Figure 2: Comparisons between the performance of Kendall's rank correlation and cosine similarity on the test set of mini-ImgeNet under 5-way 1-shot after each epoch, when pre-training a Conv-4 network from scratch.

Figure 3: Comparisons between the performance of leveraging Kendall's rank correlation for both training and testing, and for testing only.

process, Kendall's rank correlation achieves better performance compared to cosine similarity, demonstrating the effectiveness of leveraging channel importance ranking in few-shot learning.

## 5 DiffKenall: Learning with Differentiable Kendall's Rank Correlation

We have shown that the mere utilization of Kendall's rank correlation at test time yields a substantial enhancement in few-shot learning performance. In this section, we investigate the potential for further improvements through the integration of Kendall's rank correlation into the meta learning process. The main challenge is that the calculation of channel importance ranking is non-differentiable, which hinders the direct optimization of Kendall's rank correlation for training. To tackle this problem, in our study, we propose a differentiable Kendall's rank correlation by approximating Kendall's rank correlation with smooth functions, hence enabling optimizing ranking consistency directly in episodic training.

### 5.1 A Differentiable Approximation of Kendall's Rank Correlation

Given two $n$-dimensional vectors $\boldsymbol{x} = (x_1, ..., x_n)$, $\boldsymbol{y} = (y_1, ..., y_n)$, we define $\tilde{\tau}_\alpha(\boldsymbol{x}, \boldsymbol{y})$ as:

$$\tilde{\tau}_\alpha(\boldsymbol{x}, \boldsymbol{y}) = \frac{1}{N_0} \sum_{i=2}^{n} \sum_{j=1}^{i-1} \frac{e^{\alpha(x_i-x_j)} - e^{-\alpha(x_i-x_j)}}{e^{\alpha(x_i-x_j)} + e^{-\alpha(x_i-x_j)}} \frac{e^{\alpha(y_i-y_j)} - e^{-\alpha(y_i-y_j)}}{e^{\alpha(y_i-y_j)} + e^{-\alpha(y_i-y_j)}}, \tag{3}$$

where $\alpha > 0$ is a hyperparameter, and $N_0 = \frac{n(n-1)}{2}$ represents the total number of channel pairs.

**Lemma 1.** $\tilde{\tau}_\alpha(\boldsymbol{x}, \boldsymbol{y})$ *is a differentiable approximation of Kendall's rank correlation* $\tau(\boldsymbol{x}, \boldsymbol{y})$,

$$\tau(\boldsymbol{x}, \boldsymbol{y}) = \lim_{\alpha \to +\infty} \tilde{\tau}_\alpha(\boldsymbol{x}, \boldsymbol{y}).$$

Please refer to the appendix for the proof. The main idea of the proof involves using a sigmoid to approximate the sgn function in the sgn expression of Kendall's rank correlation where $\tau(\boldsymbol{x}, \boldsymbol{y}) = \frac{2}{n(n-1)} \sum_{i<j} \operatorname{sgn}(x_i - x_j) \operatorname{sgn}(y_i - y_j)$.

### 5.2 Integrating Differentiable Kendall's Rank Correlation with Meta-Baseline

By approximating Kendall's rank correlation using Eq. (3), we are able to directly optimize Kendall's rank correlation in training, addressing the issue of non-differentiability in rank computation. This implies that our method can integrate with numerous existing approaches that rely on geometric similarity, by *replacing geometric similarity with differentiable Kendall's rank correlation* in episodic training, thereby achieving further improvements.

**Algorithm 1:** Episodic training with differentiable Kendall's rank correlation $\tilde{\tau}_a(\boldsymbol{x}, \boldsymbol{y})$

---

**Input:** Base class dataset $\mathcal{D}_{\text{base}} = \{(x_i, y_i)|i = 1, ..., N\}$.
**Output:** The episodic-traing loss $\mathcal{L}$.

**1** Randomly sample $n$ categories from base class dataset $\mathcal{D}_{\text{base}}$;

**2** Randomly sample $m_s$ samples in each category to build the support set $\mathcal{S}$;

**3** Randomly sample $m_q$ samples in each category to build the query set $\mathcal{Q}$;

**4** Compute class prototypes: $c_k = \frac{1}{|\mathcal{S}_k|} \sum\limits_{(x,y)\in\mathcal{S}_k} f_\theta(x)$. $\mathcal{S}_k$ denotes the subset of $\mathcal{S}$ where $y = k$;

**5** Compute the episodic-training loss: $\mathcal{L} = -\frac{1}{|\mathcal{Q}|} \sum\limits_{(x,y)\in Q} \log p(y|x)$. $p(y|x)$ is obtained by Eq. (1),

where $\tilde{\tau}_a(\boldsymbol{x}, \boldsymbol{y})$ is used as the similarity measure $\text{sim}(\cdot)$.

---

Table 2: Comparison studies on **mini-ImageNet** and **tiered-ImageNet**. The average accuracy (%) with 95% confidence interval of the 5-way 1-shot setting and the 5-way 5-shot setting is reported.

| Dataset | Method | Backbone | 5-way 1-shot | 5-way 5-shot |
|---|---|---|---|---|
| mini-ImageNet | ProtoNet [1] | Conv-4 | $49.42 \pm 0.78$ | $68.20 \pm 0.66$ |
| | MatchingNet [23] | Conv-4 | $43.56 \pm 0.84$ | $55.31 \pm 0.73$ |
| | MAML [11] | Conv-4 | $48.70 \pm 1.84$ | $63.11 \pm 0.92$ |
| | GCR [30] | Conv-4 | $53.21 \pm 0.40$ | $72.34 \pm 0.32$ |
| | SNAIL [31] | ResNet-12 | $55.71 \pm 0.99$ | $68.88 \pm 0.92$ |
| | AdaResNet [32] | ResNet-12 | $56.88 \pm 0.62$ | $71.94 \pm 0.57$ |
| | TADAM [14] | ResNet-12 | $58.50 \pm 0.30$ | $76.70 \pm 0.30$ |
| | MTL [33] | ResNet-12 | $61.20 \pm 1.80$ | $75.50 \pm 0.80$ |
| | MetaOptNet [12] | ResNet-12 | $62.64 \pm 0.61$ | $78.63 \pm 0.46$ |
| | TapNet [34] | ResNet-12 | $61.65 \pm 0.15$ | $76.36 \pm 0.10$ |
| | CAN [3] | ResNet-12 | $63.85 \pm 0.48$ | $79.44 \pm 0.34$ |
| | ProtoNet + TRAML [35] | ResNet-12 | $60.31 \pm 0.48$ | $77.94 \pm 0.57$ |
| | SLA-AG [36] | ResNet-12 | $62.93 \pm 0.63$ | $79.63 \pm 0.47$ |
| | ConstellationNet [6] | ResNet-12 | $64.89 \pm 0.23$ | $79.95 \pm 0.17$ |
| | Meta-Baseline [4] | ResNet-12 | $63.17 \pm 0.23$ | $79.26 \pm 0.17$ |
| | Meta-Baseline + DiffKendall (Ours) | ResNet-12 | $\mathbf{65.56 \pm 0.43}$ | $\mathbf{80.79 \pm 0.31}$ |
| tiered-ImageNet | ProtoNet [1] | Conv-4 | $53.31 \pm 0.89$ | $72.69 \pm 0.74$ |
| | Relation Networks [2] | Conv-4 | $54.48 \pm 0.93$ | $71.32 \pm 0.78$ |
| | MAML [11] | Conv-4 | $51.67 \pm 1.81$ | $70.30 \pm 1.75$ |
| | MetaOptNet [12] | ResNet-12 | $65.99 \pm 0.72$ | $81.56 \pm 0.53$ |
| | CAN [3] | ResNet-12 | $69.89 \pm 0.51$ | $84.23 \pm 0.37$ |
| | Meta-Baseline [4] | ResNet-12 | $68.62 \pm 0.27$ | $83.74 \pm 0.18$ |
| | Meta-Baseline + DiffKendall (Ours) | ResNet-12 | $\mathbf{70.76 \pm 0.43}$ | $\mathbf{85.31 \pm 0.34}$ |

A straightforward application is the integration of our method with Meta-Baseline [4], which is a simple and widely-adopted baseline in few-shot learning. In Meta-Baseline, cosine similarity $\cos(\boldsymbol{x}, \boldsymbol{y})$ is employed as the similarity measure $\text{sim}(\cdot)$ in Eq. (1), to determine the semantic similarity between query samples' embedding and prototypes in episodic training and testing. Hence, we replace the cosine similarity originally utilized in Meta-Baseline with Kendall's rank correlation by employing differentiable Kendall's rank correlation during the meta-training phase and adopting Kendall's rank correlation during the testing phase. The outline of calculating the episodic-training loss with differentiable Kendall's rank correlation $\tilde{\tau}_a(\boldsymbol{x}, \boldsymbol{y})$ is demonstrated in Algorithm 1.

## 5.3 Results

**Settings.** We conduct extensive experiments on mini-ImageNet [23] and tiered-ImageNet [29] for performance evaluation, both of which are widely used in previous studies. We use ResNet-12 as the backbone network and first pre-train the feature extractor on the base dataset. $\alpha$ in Eq. (3) is set to 0.5 for the differentiable approximation of Kendall rank correlation. The performance evaluation is conducted on randomly sampled tasks from the test set, where the average accuracy and the 95% confidence interval are reported.

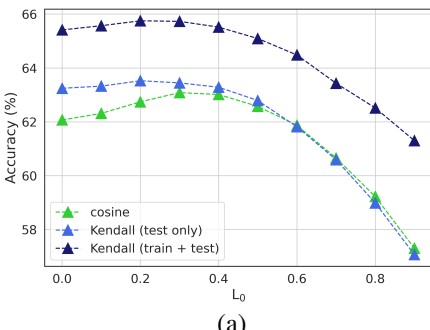 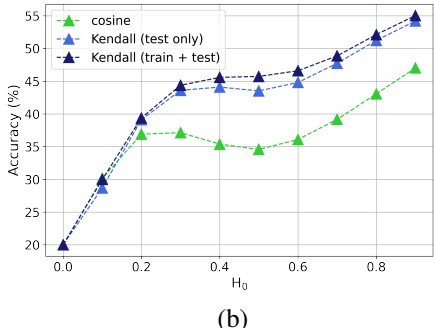

| (a) | (b) |

Figure 4: Average accuracy of 5-way 1-shot tasks on the test-set of mini-ImageNet using the masked features. (a) Channels with values less than $L_0$ are masked out (b) Channels with values larger than $H_0$ are masked out.

**Comparison Studies.** Table 2 shows the results of the comparison studies. It can be seen that on both the datasets, compared with the original Meta-Baseline that uses cosine similarity in episodic training, we achieve a clear improvement by replacing cosine similarity with the proposed differentiable Kendall's rank correlation, with $2.39\%$, $2.16\%$ in the 1-shot and $1.53\%$, $1.57\%$ in the 5-shot, respectively. Moreover, our method also outperforms methods like CAN [3] and ConstellationNet [6], where cross-attention and self-attention blocks are used. It is worth noting that there are no additional architectures or learnable parameters introduced in our method, just like the original Meta-Baseline.

Furthermore, we also conduct a comprehensive comparison to demonstrate the role of incorporating differentiable Kendall's rank correlation during episodic training. The results are presented in Figure 3. Compared with solely adopting Kendall's rank correlation at test time, it can be observed that leveraging the differentiable Kendall's rank correlation in episodic training leads to a $1\%$-$2\%$ improvement under 5-way 1-shot on test sets with varying domain discrepancies. This clearly demonstrates that the proposed differentiable Kendall's rank correlation can effectively serve as a soft approximation to directly optimize ranking consistency for further performance improvements in few-shot learning.

## 5.4 Analysis

**Channel-Wise Ablation Studies: A closer look at the performance improvements.** We aim to carry out an in-depth analysis to uncover the underlying reasons behind the performance gains observed in few-shot learning upon utilizing Kendall's rank correlation. By determining semantic similarities between features with the correlation of channel importance ranking, we can effectively distinguish the role and contribution of small-valued channels that overwhelmingly occupy the feature space of novel samples for classification. As a result, these previously neglected small-valued channels can be fully harnessed to enhance classification performance. To validate this, we propose a channel-wise ablation study in which we test the performance of models on few-shot tasks using the small-valued and large-valued channels separately, allowing for a more detailed and nuanced understanding of their respective roles in classification. Concretely, given an $n$-dimensional feature $\boldsymbol{x} = (x_1, ..., x_n)$, we define two types of masks $\boldsymbol{l} = (l_1, ..., l_n)$, $\boldsymbol{h} = (h_1, ..., h_n)$, as follows,

$$l_i = \begin{cases} 0 & \text{if } x_i < L_0 \\ 1 & \text{else} \end{cases} \qquad h_i = \begin{cases} 0 & \text{if } x_i > H_0 \\ 1 & \text{else} \end{cases}$$

The masked feature is then calculated as $\bar{\boldsymbol{x}} = \boldsymbol{x} \odot \boldsymbol{l}$ or $\bar{\boldsymbol{x}} = \boldsymbol{x} \odot \boldsymbol{h}$, where $\odot$ denotes Hadamard Product. We selectively preserve channels with large values in mask $\boldsymbol{l}$ while masking out small-valued channels. Conversely, we exclude channels with large values in mask $\boldsymbol{h}$ to exclusively evaluate the performance of small-valued channels on classification. Subsequently, we utilize the masked embedding to compare the performance of Kendall's rank correlation and cosine similarity in few-shot learning under various settings of threshold $L_0$ and $H_0$ with the corresponding results illustrated in Figure 4.

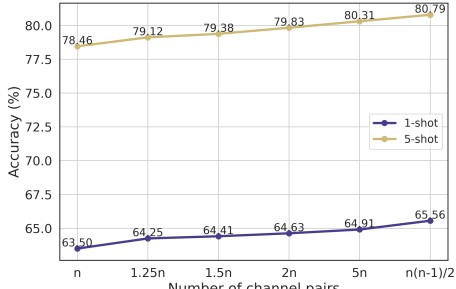
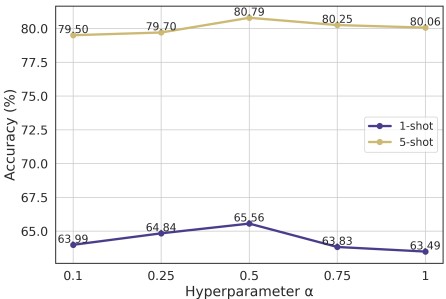

Figure 5: Performance of the linear time complexity calculation method for Kendall's rank correlation on mini-ImageNet ($n = 640$).

Figure 6: Ablation studies of the hyperparameter $\alpha$ in Eq. (3) on mini-ImageNet.

Figure 4(a) shows that when small-valued channels are masked during testing, both cosine similarity and Kendall's rank correlation achieve similar performance, but significant improvements are observed by utilizing differentiable Kendall's rank correlation for episodic training. As small-valued channels are gradually unmasked, Kendall's rank correlation significantly outperforms cosine similarity. This demonstrates that the improvement in performance achieved by utilizing Kendall's rank correlation is due to a more effective utilization of small-valued channels in novel sample features. This effect is further reflected in Figure 4(b), where masking only large-valued channels and utilizing only small-valued channels for classification results in a substantial improvement of approximately 9% in performance using Kendall's rank correlation compared to cosine similarity.

**Calculating Kendall's Rank Correlation within Linear Time Complexity.** Kendall's rank correlation requires us to compute the importance ranking concordance of any pair of channels. This results in a higher time complexity compared to cosine similarity, increasing quadratically with the total number of channels. We investigate whether this time complexity could be further reduced to improve the computational efficiency of Kendall's rank correlation at test time. Specifically, we propose a simple approach to calculate the importance ranking concordance by randomly sampling a subset of channel pairs instead of using all channel pairs. The experimental results are presented in Figure 5, where $n$ represents the total number of channels in the features of novel samples. It can be observed that by randomly sampling $5n$ channel pairs, we achieve a performance that is very close to using all channel pairs. It should be noted that this performance has already surpassed that of the original Meta-Baseline method while the time complexity is maintained linear, equivalent to cosine similarity.

**Hyperparameter Sensitivity.** We also investigate the impact of the hyperparameter $\alpha$ in Eq. (3), and the experimental results are presented in Figure 6. The best results are obtained around a value of $0.5$, and the performance is found to be relatively insensitive to variations of $\alpha$ within a certain range. Setting a value for $\alpha$ that is too large or too small may lead to a decrease in performance. When a value for $\alpha$ is too large, the model may overfit to the base classes during episodic training, which can result in decreased generalization performance on novel classes. Conversely, if a value that is too small is used, this may lead to a poor approximation of Kendall's rank correlation.

## 5.5 Visualization

Further visual analysis is demonstrated in Figure 7. Specifically, we employ Kendall's rank correlation and cosine similarity to visualize the feature maps of the query samples. By computing the semantic similarity between the class prototype and each local feature of the query samples, the regions wherein the salient targets are located on the feature maps are illustrated. Hence, we can observe whether the significant features in the query samples are accurately detected. It is evident that the utilization of Kendall's rank correlation results in a more precise localization of the distinctive regions within the query sample.

Furthermore, we conduct in-depth visual experiments involving channel ablation. We mask the channels with values greater than $H_0$ in the features of both the class prototype and query sample,

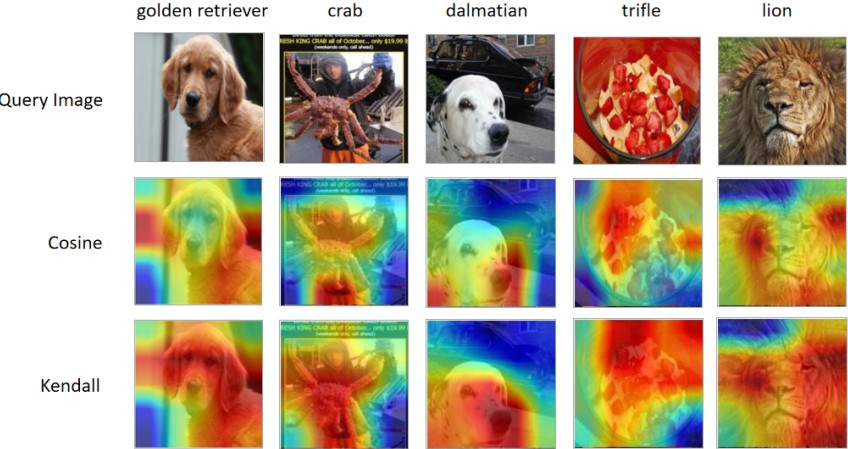

Figure 7: The results of the visual analysis on the test set of mini-ImageNet with cosine similarity and Kendall's Rank Correlation respectively. It can be seen that Kendall's rank correlation demonstrates a more accurate localization of salient targets and superior differentiation of key features.

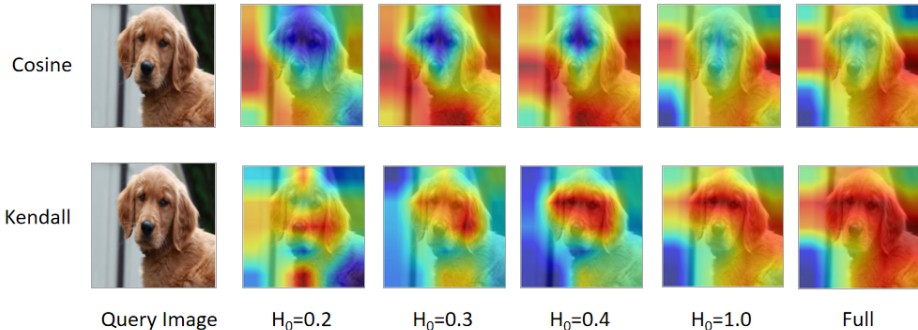

Figure 8: The results of the channel ablation visualization experiments, where channels with values greater than $H_0$ are masked out, and 'Full' indicates when all channels are used. It can be seen that the key features of the category are held on the small-valued channels and are successfully uncovered by Kendall's rank correlation, while cosine similarity misses these critical features.

just like the channel-wise ablation experiments in Section 5.4. The results are shown in Figure 8, from which we can observe that Kendall's rank correlation captures the discriminative features in the query sample when only utilizing the small-valued channels. In contrast, cosine similarity ignores these critical features, resulting in an inability to correctly locate salient regions when all channels are used. Therefore, we can infer that the small-valued channels that occupy the majority of the features indeed play a vital role in few-shot learning. This also explicitly demonstrates that the improvement achieved by Kendall's rank correlation in few-shot learning is essentially due to its ability to fully exploit the small-valued channels in features.

## 6 Conclusion

This paper exposes a key property of the features of novel samples in few-shot learning, resulting from the fact that values on most channels are small and closely distributed, making it arduous to distinguish their importance in classification. To overcome this, we propose to replace the commonly used geometric similarity metric with the correlation of the channel importance ranking to determine semantic similarities. Our method can integrate with numerous existing few-shot approaches without increasing training costs and has the potential to integrate with future state-of-the-art methods that rely on geometric similarity metrics to achieve additional improvement.

## Acknowledgment

This work is supported by 2023 CCF-Baidu Open Fund and Microsoft Research Asia.

We would like to express our sincere gratitude to the reviewers of NeurIPS 2023 for their insightful and constructive feedback. Their valuable comments have greatly contributed to improving the quality of our work.

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

# Appendix

## A   Proof for Lemma 1

**Lemma 1.** $\tilde{\tau}_\alpha(\boldsymbol{x}, \boldsymbol{y})$ *is a differentiable approximation of Kendall's rank correlation $\tau(\boldsymbol{x}, \boldsymbol{y})$,*

$$\tau(\boldsymbol{x}, \boldsymbol{y}) = \lim_{\alpha \to +\infty} \tilde{\tau}_\alpha(\boldsymbol{x}, \boldsymbol{y}).$$

*Proof.* First, consider the scenario where channel pairs exhibit consistent importance ranking, specifically, either $x_i > x_j \wedge y_i > y_j$ or $x_i < x_j \wedge y_i < y_j$. In the case where $x_i > x_j \wedge y_i > y_j$, we obtain:

$$\lim_{\alpha \to +\infty} \frac{e^{\alpha(x_i - x_j)} - e^{-\alpha(x_i - x_j)}}{e^{\alpha(x_i - x_j)} + e^{-\alpha(x_i - x_j)}} = 1, \quad \lim_{\alpha \to +\infty} \frac{e^{\alpha(y_i - y_j)} - e^{-\alpha(y_i - y_j)}}{e^{\alpha(y_i - y_j)} + e^{-\alpha(y_i - y_j)}} = 1.$$

On the other hand, if $x_i < x_j \wedge y_i < y_j$, we have:

$$\lim_{\alpha \to +\infty} \frac{e^{\alpha(x_i - x_j)} - e^{-\alpha(x_i - x_j)}}{e^{\alpha(x_i - x_j)} + e^{-\alpha(x_i - x_j)}} = -1, \quad \lim_{\alpha \to +\infty} \frac{e^{\alpha(y_i - y_j)} - e^{-\alpha(y_i - y_j)}}{e^{\alpha(y_i - y_j)} + e^{-\alpha(y_i - y_j)}} = -1.$$

Hence, when $x_i > x_j \wedge y_i > y_j$ or $x_i < x_j \wedge y_i < y_j$, the following conclusion holds:

$$\lim_{\alpha \to +\infty} \frac{e^{\alpha(x_i - x_j)} - e^{-\alpha(x_i - x_j)}}{e^{\alpha(x_i - x_j)} + e^{-\alpha(x_i - x_j)}} \frac{e^{\alpha(y_i - y_j)} - e^{-\alpha(y_i - y_j)}}{e^{\alpha(y_i - y_j)} + e^{-\alpha(y_i - y_j)}} = 1. \tag{4}$$

Second, consider the scenario where channel pairs exhibit inconsistent importance ranking, that is, either $x_i > x_j \wedge y_i < y_j$ or $x_i < x_j \wedge y_i > y_j$. In the case where $x_i > x_j \wedge y_i < y_j$, we obtain:

$$\lim_{\alpha \to +\infty} \frac{e^{\alpha(x_i - x_j)} - e^{-\alpha(x_i - x_j)}}{e^{\alpha(x_i - x_j)} + e^{-\alpha(x_i - x_j)}} = 1, \quad \lim_{\alpha \to +\infty} \frac{e^{\alpha(y_i - y_j)} - e^{-\alpha(y_i - y_j)}}{e^{\alpha(y_i - y_j)} + e^{-\alpha(y_i - y_j)}} = -1.$$

On the other hand, if $x_i < x_j \wedge y_i > y_j$, we have:

$$\lim_{\alpha \to +\infty} \frac{e^{\alpha(x_i - x_j)} - e^{-\alpha(x_i - x_j)}}{e^{\alpha(x_i - x_j)} + e^{-\alpha(x_i - x_j)}} = -1, \quad \lim_{\alpha \to +\infty} \frac{e^{\alpha(y_i - y_j)} - e^{-\alpha(y_i - y_j)}}{e^{\alpha(y_i - y_j)} + e^{-\alpha(y_i - y_j)}} = 1.$$

Hence, when $x_i > x_j \wedge y_i < y_j$ or $x_i < x_j \wedge y_i > y_j$, the following conclusion holds:

$$\lim_{\alpha \to +\infty} \frac{e^{\alpha(x_i - x_j)} - e^{-\alpha(x_i - x_j)}}{e^{\alpha(x_i - x_j)} + e^{-\alpha(x_i - x_j)}} \frac{e^{\alpha(y_i - y_j)} - e^{-\alpha(y_i - y_j)}}{e^{\alpha(y_i - y_j)} + e^{-\alpha(y_i - y_j)}} = -1. \tag{5}$$

When considering all channel pairs, combining Eq. (4) and Eq. (5), it is evident that:

$$\lim_{\alpha \to +\infty} \tilde{\tau}_\alpha(\boldsymbol{x}, \boldsymbol{y}) = \lim_{\alpha \to +\infty} \frac{1}{N_0} \sum_{i=2}^{n} \sum_{j=1}^{i-1} \frac{e^{\alpha(x_i - x_j)} - e^{-\alpha(x_i - x_j)}}{e^{\alpha(x_i - x_j)} + e^{-\alpha(x_i - x_j)}} \frac{e^{\alpha(y_i - y_j)} - e^{-\alpha(y_i - y_j)}}{e^{\alpha(y_i - y_j)} + e^{-\alpha(y_i - y_j)}}$$

$$= \frac{N_{\mathrm{con}} - N_{\mathrm{dis}}}{N_0}$$

$$= \tau(\boldsymbol{x}, \boldsymbol{y}),$$

where $N_{\mathrm{con}}$ represents the total count of channel pairs with consistent importance ranking, $N_{\mathrm{dis}}$ represents the count of channel pairs with inconsistent importance ranking.  $\square$

