# OpenReview forum: "DiffKendall: A Novel Approach for Few-Shot Learning with Differentiable Kendall's Rank Correlation"
_NeurIPS.cc/2023/Conference — NeurIPS 2023 poster_

### Official Review · Reviewer_DzME · 2023-07-03

**Soundness:** 3 good
**Presentation:** 3 good
**Contribution:** 2 fair
**Rating:** 4
**Confidence:** 4

**Summary:**

This paper is motivated by the observation that the value of channels in novel dataset has a much more uniform distribution than that in the base dataset.
Based on the observation, the author proposes a new similarity metric, Kendall’s rank correlation, to utilize the ranking information of channels instead of values to calculate the similarity of two embeddings.
To address the non-differentiability issue of Kendall’s rank correlation, the author also proposes a differentiable approximation for meta-learning.
Experiments show that the Kendall’s rank correlation and proposed approximation Kendall’s rank correlation are useful in some cases.


**Strengths:**

1.	The idea that uses the ranking information of channels instead of values to calculate the similarity of two embeddings is interesting.
2.	The analysis in Section 5.4 is helpful to understand how the magnitude of channels’ value affects the performance at test-time.


**Weaknesses:**

1.	The proposed Kendall’s rank correlation is constrained with FSL methods that uses similarity metric to determine the similarity between two images, which limits the application scope of it.
2.	The comparison in Section 4.2 is not convincing. The authors only consider a baseline that uses cosine similarity metric, which is not a very popular design for many FSL methods. Compared with more baselines using other similarity metrics (like Square Euclidean Distance in ProtoNet and a learnable metric in RelationNet) may help me to make sure the effectiveness and generality of proposed Kendall’s rank correlation.
3.	The reported results in Table 1 are not clear. For example, why the results of baseline (cos+CE) and of CIM is much lower than that in Table 1 of [1] (lower than 10~20 points)?
4.	The comparison methods (e.g. CAN and ConstellationNet) in Table 2 and 3 are not state-of-the-art methods as the paper claimed. Compared with more recent FSL methods (e.g. [2],[3],[4]) maybe more convincing to support the opinion that proposed method outperforms the state-of-the-art methods.

[1] Channel importance matters in few-shot image classification. ICML 2022.
[2] Alleviating the Sample Selection Bias in Few-shot Learning by Removing Projection to the Centroid. NeurIPS 2022.
[3] Improving Task-Specific Generalization in Few-Shot Learning via Adaptive Vicinal Risk Minimization. NeurIPS 2022.
[4] Matching Feature Sets for Few-Shot Image Classification. CVPR 2022.


**Questions:**

Please refer to "weaknesses" part for all my concerns.

**Limitations:**

Overall, this paper has limited contribution and unconvincing experimental results.

---

> ### Author Rebuttal · Authors · 2023-08-09
>
> Q1: The proposed Kendall’s rank correlation is constrained with FSL methods that uses similarity metric to determine the similarity between two images, which limits the application scope of it.
>
> A1: Thank you for your feedback. In fact, research in few-shot learning can be broadly categorized into metric-learning-based and optimization-based methods, with **metric-learning-based methods constituting a significant portion**. **Similarity measurement is an essential component of metric-learning-based approaches**. Our proposed method holds the potential to seamlessly integrate with this category by replacing cosine similarity with Kendall's rank correlation. On the other hand, as our method leverages Kendall's Rank Correlation to achieve consistency across feature channels, it can be easily integrated with those methods that address few-shot learning from other perspectives.
>
> Moreover, **the applicability of our method extends beyond the realm of few-shot learning**, as its underlying motivation is from observing an apparent difference in feature channel values between base data and novel data. We found that compared to base classes, when the feature extractor faces a novel class that is unseen before, most features' channels have small and closely clustered values, making it difficult for the model to distinguish the importance of individual channels. At this point, channel importance ranking can effectively accentuate the differences between channels. This property is applicable not only to few-shot learning but also to any task with cross-domain generalization attributes and holds inspiration.
>
> Based on the above aspects, we are confident that our method enjoys a broad application scope, far beyond the limitations initially perceived.
>
> Q2: The comparison in Section 4.2 is not convincing. The authors only consider a baseline that uses cosine similarity metric, which is not a very popular design for many FSL methods. Compared with more baselines using other similarity metrics (like Square Euclidean Distance in ProtoNet and a learnable metric in RelationNet) may help me to make sure the effectiveness and generality of proposed Kendall’s rank correlation.
>
> A2: Thanks for the comments. The **learnable metric-based methods** require meta-learning to train and obtain the optimal parameter values. However, in Section 4.2, we aimed to demonstrate that our approach does not require any training and solely relies on using Kendall rank correlation in the inference phase to achieve noticeable performance improvement. Additionally, the reason why we did not include the **Euclidean distance** in our experiments is that on the unit sphere, cosine similarity and Euclidean distance are equivalent. Furthermore, in few-shot learning, only ProtoNet adopts Euclidean distance, while many subsequent works are based on cosine similarity and have shown that using cosine similarity generally yields better results compared to using Euclidean distance.
>
> Nonetheless, we also **conduct experiments using the learnable metric in RelationNet and Euclidean distance** (See Table B of the attached PDF in our "global" response). The experimental results demonstrate the superiority of utilizing Kendall's rank correlation over the learnable metric in RelationNet and Euclidean distance.
>
> Q3: The reported results in Table 1 are not clear. For example, why the results of baseline (cos+CE) and of CIM is much lower than that in Table 1 of [1] (lower than 10~20 points)?
>
> A3: Sorry for any confusion caused. In fact, CIM’s Table 1 reports results for the **5-way 5-shot** setting, while our Table 1 presents results for the **5-way 1-shot** setting. We also conduct experiments in the 5-way 5-shot setting (See Table D of the attached PDF in our "global" response). As you can see, in fact, our reimplementation outperforms the results reported in CIM's Table 1, and **the effectiveness of Kendall's rank correlation is demonstrated**.
>
> Q4: The comparison methods (e.g. CAN and ConstellationNet) in Table 2 and 3 are not state-of-the-art methods as the paper claimed. Compared with more recent FSL methods (e.g. [2],[3],[4]) maybe more convincing to support the opinion that proposed method outperforms the state-of-the-art methods.
>
> A4: Thank you for bringing this to our attention. We will ensure the inclusion of the methods you've highlighted in our list of citations. It is pertinent to highlight that our latest experimental results encompass comparisons with more recent FSL methods, including those you've referenced (Please refer to Table A in the PDF in our "global" response).
> We would like to highlight that by integrating Kendall's Rank Correlation into a stronger backbone DeepEMD, **our method achieves SOTA performance**.

---

> ### Author Response · Authors · 2023-08-15
> **We would be grateful if you could take a look at the response**
>
> Dear Reviewer DzME:
>
> We sincerely appreciate your valuable time devoted to reviewing our manuscript. We would like to gently remind you of the **approaching deadline for the discussion phase**. We have diligently addressed the issues you raised in your feedback, providing detailed explanations. For instance, we have addressed your concerns regarding the reliability of our experimental results. Moreover, to comprehensively showcase the superiority of Kendall rank correlation, we have included comparisons between Kendall's rank correlation, Euclidean distance, and learnable distances. Furthermore, We have also included comparative experiments with SOTA methods, including the methods mentioned in your citations, demonstrating that by straightforwardly replacing cosine similarity with Kendall’s rank correlation, our method achieves state-of-the-art performance when combined with a stronger baseline DeepEMD. Would you kindly take a moment to look at it?
>
> We are very enthusiastic about engaging in more in-depth discussions with you.

---

> > ### Comment · Reviewer_DzME · 2023-08-16
> > **Thanks for the response**
> >
> > Dear authors,
> >
> > I have carefully read your rebuttal including the new experimental results. Some of my concerns (e.g., the confusion about the results in Table 1) have been properly addressed. Additional results are provided to show the effectiveness of your method. However, my doubts on the novelty and performance still exist. Specifically, your method with strong backbone DeepEMD (which is already SOTA) gets marginal improvement.  Thanks for your detailed response, but I would like to keep my rating.

---

> > > ### Author Response · Authors · 2023-08-16
> > >
> > > Thank you for your valuable feedback. **We humbly acknowledge that our previous rebuttal addressed some of your concerns.** Regarding the points you highlighted about novelty and performance, we hope to offer a more thorough discussion on this matter.
> > > 1. As you mentioned, DeepEMD already exhibits a remarkable level of performance. **Further improvement on such a strong baseline is no trivial task; in fact, it poses a considerable challenge.** The stronger the baseline, the greater the challenge. Nevertheless, we would like to highlight that **our simple modification to DeepEMD**, the replacement of the cosine similarity with Kendall’s Rank correlation, **led to a notable improvement of 1.41%** on the mini-ImageNet (68.09% -> 69.50%) and **1.94%** on the tiered-ImageNet (71.16% -> 73.10%) in 1-shot setting. **This emphasizes both the simplicity and effectiveness of our approach.** As an example for your reference, FRN (accepted by CVPR) also involves direct modifications to DeepEMD, but yielded a modest improvement of **merely 0.54%** on the mini-ImageNet.
> > > 2. Additionally, **for most recently proposed methods** such as DeepEMD, CIM and InfoPatch, integrating our method with them can **obtain consistent improvement** across **various datasets with domain differences** as shown in Table 1 and Table A, even simply employing Kendall's rank correlation during the inference stage. This suggests that **our method is ready for integration with future SOTA methods**, paving the way for further enhancements and maintaining its leading-edge status.
> > > 3. Moreover, one of significant importance is the **strong generality of our approach**. Our method possesses the capability to seamlessly integrate with various cosine-based methods, and it also **holds the potential for extension into other domains**.
> > > 4. Furthermore, we would like to emphasize that **the motivation behind our proposed method is an unexplored aspect in prior research**. Our approach brings to light a novel observation within few-shot learning -- namely, the revelation that for a novel-class, features’ channels possess smaller and closely clustered values when compared to base-classes. We have substantiated this as a universally valid inference. This situation creates a challenge in employing geometric similarity to accurately distinguish the importance among feature channels. The utilization of channel importance ranking, instead, offers an effective solution to this challenge. **All of these aspects remain unexplored in prior research.**
> > >
> > > The points we've discussed above **underscore the novelty and effectiveness of our method**. We genuinely believe in the potential and merit of our approach.
> > >
> > > Given that we have addressed part of your concerns and further elaborated on the remaining ones in this response, **we would like to humbly request a reconsideration of the scoring**. We believe our research can provide valuable insights and contributions that would be of great interest to the NeurIPS community.

---

### Official Review · Reviewer_zjGF · 2023-07-03

**Soundness:** 3 good
**Presentation:** 3 good
**Contribution:** 3 good
**Rating:** 5
**Confidence:** 5

**Summary:**

This paper proposes a new similarity metric for few-shot learning, named Kendall’s rank correlation, which originally come from the statistical concept of Kendall's rank. The motivation of this paper is from an experiment that the corresponding channel value for base and novel classes have a distribution difference. Additionally, the authors design a differentiable approximation for Kendall’s rank correlation and demonstrate favorable experimental results in comparative experiments.

**Strengths:**

1.	The paper is well-written and easy to follow, with a clear and well-organized structure.
2.	The method is straightforward and simple.


**Weaknesses:**

1.	The ablation experiments only employ cosine distance, lacking a comprehensive consideration of Euclidean distance.
2.	The performance of the differentiable approximation for various hyperparameter values is not adequately analyzed or visualized.
3.	The optimal value for the hyperparameter is stated as 0.5 but lacks experimental or theoretical evidence to support this claim.
4.	Table 3 includes the "meta-baseline" method, which belongs to another approach and should not be classified as part of the proposed method.
5.	The testing procedure is not described at all, including whether the differentiable approximation or Kendall's rank is used during testing.
6.	The reproducibility of the source code is not well identified.


**Questions:**

Please see the Weaknesses.

**Limitations:**

The primary concern lies in the reproducibility of the source code, as it is currently non-functional.

---

> ### Author Rebuttal · Authors · 2023-08-09
>
> Q1: The ablation experiments only employ cosine distance, lacking a comprehensive consideration of Euclidean distance.
>
> A1: Thank you for your feedback. We would like to point out that the reason why we did not include the Euclidean distance in our experiments is that on the unit sphere, cosine similarity and Euclidean distance are equivalent. Furthermore, in few-shot learning, only ProtoNet adopts Euclidean distance, while many subsequent works are based on cosine similarity and have shown that using cosine similarity generally yields better results compared to using Euclidean distance. Nonetheless, we have also included comparative experiments with Euclidean distance (See Table B of the attached PDF in our "global" response). The experimental results demonstrate the superiority of utilizing Kendall's rank correlation over Euclidean distance.
>
> Q2:The performance of the differentiable approximation for various hyperparameter values is not adequately analyzed or visualized.
>
> A2: Thank you for this feedback. We would like to point out that the performance of differentiable Kendall ranking correlation shows relatively low sensitivity to variations in the hyperparameter $\alpha$ within a specific range. Our experimental findings suggest that setting alpha to approximately 0.5 yields favorable performance. Setting this parameter too high may lead to overfitting on the base class data, while setting it too low may result in inadequate approximation to Kendall ranking correlation. A relevant analysis of this phenomenon has been carried out in the ablation experiments presented in Section 5.4. Here, we also conducted more detailed ablation experiments concerning this hyperparameter as follow.
>
> | Method| $\alpha$=0.1 | $\alpha$=0.2 |$\alpha$=0.3 |$\alpha$=0.4 |   $\alpha$=0.5 | $\alpha$=0.6 |$\alpha$=0.7 |$\alpha$=0.8 |
> |:---:|:---:|:---: |:---:|:---:|:---:|:---: |:---:|:---:|
> |Kendall| 63.99|64.66|64.92|65.21|65.56|65.02|64.56|63.79|
>
> Q3: The optimal value for the hyperparameter is stated as 0.5 but lacks experimental or theoretical evidence to support this claim.
>
> A3: Thank you for this feedback. In Section 5.4 and in response to Q2, we have extensively investigated this hyperparameter through ablation experiments. We did not meticulously tune this hyperparameter to deliberately seek a better result. What we discovered from the ablation experiments is that setting $\alpha$ around 0.5 yields a relatively favorable outcome.
>
> Q4: Table 3 includes the "meta-baseline" method, which belongs to another approach and should not be classified as part of the proposed method.
>
> A4: Sorry for any confusion caused. In fact, Meta-Baseline is a a simple and effective method in few-shot learning, which proposes a two-stage training paradigm. Specifically, the model is first pre-trained on the base dataset using cross-entropy loss, following conventional supervised learning. During the meta-training stage, tasks are sampled from the base dataset, simulating the construction of test tasks in the form of N-way K-shot. The training objective is to accurately classify the query samples from the sampled tasks using cross-entropy loss as the loss function. In Meta-Baseline, cosine similarity is employed as the similarity measure to determine the semantic similarity between the query samples' embeddings and prototypes for nearest-neighbor classification. Our proposed method simply replaces the cosine similarity used in Meta-Baseline with the differentiable Kendall similarity for episodic training while keeping all other settings consistent. The relevant details can be found in Section 5.2 of the paper.
>
> Q5: The testing procedure is not described at all, including whether the differentiable approximation or Kendall's rank is used during testing.
>
> A5: Sorry for any confusion caused. In fact, in the **implementation details** provided in Section 5.2, we thoroughly describe our testing process: In the testing phase, we employ Kendall's rank correlation to compute the similarity between the embeddings of query samples and class prototypes for nearest-neighbor classification. Performance evaluation is conducted on 2000 randomly sampled tasks from the test set, and the average accuracy along with the 95% confidence interval are reported. The purpose of proposing the differentiable Kendall correlation is to address the non-differentiable issue of the original Kendall rank correlation in the ranking computation. During the testing phase, we use the original Kendall’s rank correlation for evaluation.
>
> Q6: The reproducibility of the source code is not well identified.
>
> A6: Sorry for any confusion caused. We would like to provide a detailed explanation of the content in our submitted code. The code for the inference phase is included in the 'eval.py' file, where you have the flexibility to set the testing mode. Specifically, setting the mode to 'kendall_test' implies the usage of Kendall ranking correlation in the inference phase. Regarding the training phase, it is divided into two parts, 'train_pretrain.py' and 'train_meta.py'. 'train_pretrain.py' corresponds to the pretraining process using the cross-entropy loss function. On the other hand, 'train_meta.py' is responsible for the meta-training stage, where the training mode is set to 'kendall_meta', indicating the utilization of our proposed differentiable Kendall ranking correlation for meta-training. The implementation of both the original Kendall ranking correlation and our proposed differentiable Kendall ranking correlation for meta-training can be found in the 'Models/models/kendall_fsl.py' file within the code.

---

### Official Review · Reviewer_oKqH · 2023-07-04

**Soundness:** 2 fair
**Presentation:** 3 good
**Contribution:** 2 fair
**Rating:** 3
**Confidence:** 5

**Summary:**

This paper introduces a novel approach for few-shot learning using Kendall's Rank Correlation. The authors demonstrate that feature channel importance ranking is a more reliable indicator for few-shot learning than geometric similarity metrics. They propose replacing the geometric similarity metric with Kendall's rank correlation for inference, which improves the performance of few-shot learning across different datasets and domains. Additionally, the paper presents a carefully designed differentiable loss for meta-training to address the non-differentiability issue of Kendall's rank correlation. The contributions of this paper can be summarized as follows:
1. Introducing the use of feature channel importance ranking for few-shot learning.
2. Demonstrating the effectiveness of Kendall's rank correlation in improving few-shot learning performance.
3. Proposing a differentiable approximation of Kendall's rank correlation for meta-training, leading to further performance improvements.


**Strengths:**

1. Originality: The paper presents a novel approach for few-shot learning that uses Kendall's rank correlation. This is a unique and innovative idea that has not been explored in previous research.

2. Quality: The paper is well-researched and presents a thorough analysis of the proposed method. The authors provide detailed experimental results and ablation studies to validate their approach. The proposed differentiable loss function is carefully designed and addresses the non-differentiability issue of Kendall's rank correlation.

3. Clarity: The paper is well-written and easy to understand. The paper is also well-organized, making it easy to follow the flow of ideas.

4. Significance: The use of Kendall's rank correlation has shown to be effective in improving the performance of few-shot learning across different datasets and domains.


**Weaknesses:**

1. Lack of comparison with state-of-the-art methods: The paper does not compare the proposed method with state-of-the-art few-shot learning methods.

2. Lack of theoretical justification: The paper does not provide a theoretical justification for why feature channel importance ranking and Kendall's rank correlation are better suited for few-shot learning than geometric similarity metrics. Providing such a justification would strengthen the paper's argument and make it more convincing.

3. The motivation described in the paper is not clearly explained：Why replace geometric similarity metrics with Kendall's rank correlation?

4. From the results in Table 2, it can be seen that the performance improvement is marginal, and the final result is not SOTA.


**Questions:**

1. “This often leads to a relatively uniform distribution of values across feature channels on novel classes, making it difficult to determine channel importance for novel tasks.” I can't understand the meaning of this sentence? Please provide a detailed explanation.

2. “When we compare the values of different feature channels on the base dataset and novel dataset, we observe that the novel dataset has a much more uniform value distribution than the base dataset. ” Can this conclusion hold true? This is only an observation on one dataset, and there is no evidence to suggest that the new dataset has a multiple more uniform value distribution than the base dataset.

3. The third paragraph in the introduction uses an example of dogs and wolves to illustrate that cosine similarity cannot effectively distinguish between dogs and wolves, while Figure 1 (b) shows the identification results of dogs and crabs.

4. In Table 1, for deeper backbones, why does the Kendall's rank correlation have a decrease in performance compared to cosine distance?


**Limitations:**

The authors do not provide limitations and potential negative societal impact of their work.

---

> ### Author Rebuttal · Authors · 2023-08-06
>
> Q1: Lack of comparisons with SOTA methods.
>
> A1: Thanks for the suggestion. In our latest experimental results, we have added comparisons with SOTA methods (See Table A of the PDF in our "global" response). We would like to highlight that by integrating Kendall's Rank Correlation into a stronger backbone DeepEMD, **our method achieves SOTA performance**.
>
> Q2: Lack of theoretical justification for why Kendall's rank correlation are better suited for FSL.
>
> A2: Thanks. Our method is inspired by empirical observations, making it an algorithmic paper rather than a theoretical one. Extensive experiments support our intuitive observations, which align with our expectations.
> While we will further provide intuitive justification in our next response, we leave the theoretical one as our future work.
>
> Q3: The motivation behind replacing geometric similarity metrics with Kendall's rank correlation is not clearly explained.
>
> A3: It seems that our original explanation of the motivation might not have been clear enough. Let us provide a detailed explanation below.
>
> Our approach emerged from observing an apparent difference in feature channel values between base data and novel data. We found that compared to base classes, when the feature extractor faces a novel class that is unseen before, the feature channel values become more uniform, i.e., **for a novel class, most non-core features' channels have small and closely clustered values** in the range [0.25, 0.5] (see Figure 1 of the paper). This phenomenon occurs because the model is trained on the base data, and consequently exhibits reduced variation of feature values when dealing with novel data. This situation creates a challenge in employing geometric similarity to accurately distinguish the importance among non-core feature channels.
>
> To provide a concrete example, consider distinguishing between dogs and wolves. **While they share nearly identical core visual features, minor features play a vital role in differentiating them**. Suppose the core feature, two minor features are represented by channels 1, 2, and 3, respectively, in the feature vector. A dog prototype may have feature (1, 0.3, 0.2), and a wolf prototype may have feature (1, 0.2, 0.3). Now, for a test image with feature (0.9, 0.28, 0.22), it appears more dog-like, as the 2nd feature is more prominent than the 3rd. However, cosine distance misleadingly places this test image closer to the wolf prototype (distance=0.031) rather than the dog prototype (distance=0.048). Contrastingly, the test image shares the same channel ranking (1, 2, 3) as the dog prototype, whereas the wolf prototype's channel ranking is (1, 3, 2). Inpired by this, we employ Kendall’s rank correlation to more accurately discern between dogs and wolves, highlighting the utility of our approach.
>
> We hope this clarification better conveys the underlying rationale for our method, and we will carefully review this section in the revised paper to ensure that the motivation is articulated more clearly.
>
> Q4: The results in Table 2 show marginal improvements, and the final result is not SOTA.
>
> A4: It's important to clarify that our current experiments are conducted based on a simple and widely-adopted baseline (meta-baseline). By simply substituting cosine similarity with Kendall's rank correlation, we achieve an improvement of 2%. Moreover, as mentioned in the response to Q1, our method could be easily integrated with existing methods. Combining our method with a stronger baseline, DeepEMD, we can achieve the current SOTA performance.
>
> Q5: Can't understand the meaning of the sentence mentioned.
>
> A5: In the response to Q3, we have detailed the motivation behind our approach, which is related to this question. To further clarify, let's consider an extreme scenario where all feature values are nearly equal. In this case, a minor perturbation in a feature vector could lead to significant changes in the channel ranking, while the geometric distance to other features might remain largely unchanged.
>
> Q6: Can the conclusion drawn from Figure 1 hold true?
>
> A6: We believe this is a universally valid observation. Intuitively, the model is trained on base data and hence the learned feature extractors should capture the largest variation features within that base data. For novel data that belongs to different classes from base data, the extracted features will exhibit less variation. Empirically, we additionally compare the variance of feature channel values between the base dataset and various novel datasets (See Table C). The results reveal a significantly smaller variance in feature channel values in the novel datasets compared to the base dataset. A smaller variance means values are closer to each other, which aligns with the observations depicted in Figure 1(a), validating the correctness of this conclusion.
>
> Q7: The third paragraph uses an example of dogs and wolves, while Figure 1 shows dogs and crabs.
>
> A7: The reason we use wolves and dogs as examples is that it is an intuitive idea. However, in the existing commonly used few-shot learning datasets, there are no classes that include both wolves and dogs. In our subsequent work, we will make examples and figures consistent.
>
> Q8: In Table 1, for deeper backbones, why does the Kendall's rank correlation have a decrease in performance?
>
> A8: Actually Table 1 serves as an exploratory experiment where we solely adopt Kendall’s rank correlation at test time, without any episodic training. Expecting any model to achieve performance improvement without training by simply replacing cosine similarity at test time, would be impractical. In fact, the results in Table 1 show that, in the vast majority of cases, this simple replacement leads to significant improvements. Despite the slight performance decrease on the mini-test, it is evident that across the other five datasets with more substantial domain variations, the use of Kendall’s rank correlation yields significant performance gains.

---

> > ### Comment · Reviewer_oKqH · 2023-08-17
> >
> > 1. Although integrating Kendall's Rank Correlation into a stronger backbone DeepEMD, the performance is not SOTA.
> > 2. The authors didn’t explain clearly why does the Kendall's rank correlation have a decrease in performance for deeper backbones? In the response, the authors claimed expecting any model to achieve performance improvement without training by simply replacing cosine similarity at test time, would be impractical. I think there is a problem with the expression of this sentence. In fact, if the proposed similarity measure is effective, there will be a performance improvement for various backbones. This has nothing to do with the depth of the backbone.
> > 3. I think the author still hasn't explained the motivation of the paper well, and more of it is the phenomenon observed in the experiment, without clarifying the reason behind it. If the authors can theoretically explain why Kendall's Rank Correlation can be used to replace cosine distance, it will greatly improve the quality of the paper. I think this is the core innovation of the paper.
> > In general, the rebuttal author provides partially addressed my concern. However, given the problem addressed above, I prefer not to change the rating.

---

> > > ### Author Response · Authors · 2023-08-19
> > > **Further Clarifications (1/3)**
> > >
> > > Thank you for taking the time to review our manuscript and provide your feedback. We appreciate your insights as they offer an opportunity for us to refine and clarify our work.
> > >
> > > ---
> > >
> > > *Q1. Although integrating Kendall's Rank Correlation into a stronger backbone DeepEMD, the performance is not SOTA.*
> > >
> > > We'd like to emphasize that **our method has indeed achieved SOTA results in the 1-shot setting for both mini-ImageNet and tiered-ImageNet datasets**. In the 5-shot setting, our approach maintains competitive performance relative to the current SOTA benchmarks. While we acknowledge your emphasis on SOTA performance, it's important to note that **the contribution and novelty of our work lie beyond this singular metric**.
> > >
> > > A central strength of our approach is its **simplicity** and **broad generality**. We wish to emphasize that our method can seamlessly integrate into a variety of cosine-based methods without imposing additional training costs, and it also holds the potential for extension beyond few-shot problems. As shown in Table 1 and Table A, for most recently proposed methods such as DeepEMD, CIM and InfoPatch, integrating our method with them can obtain consistent improvement across various datasets with domain differences, even simply employing Kendall's rank correlation during the inference stage. On the other hand, **while TCPR is the only method that marginally exceeds ours in the 5-shot setting, its complexity cannot be overlooked**. For a few-shot learning task, it requires hundreds or even thousands of re-samplings for data augmentation, leading to **a substantial increase in training time and effort**.
> > >
> > > Finally, we would like to kindly remind the reviewer that **top conferences have clearly stated in their reviewer guidelines that SOTA does not determine the merit or contribution of a work**.
> > >
> > > [NeurIPS](https://nips.cc/Conferences/2020/PaperInformation/ReviewerGuidelines#:~:text=Solid,%20technical%20papers%20that%20explore%20new%20territory%20or%20point%20out%20new%20directions%20for%20research%20are%20preferable%20to%20papers%20that%20advance%20the%20state%20of%20the%20art) says:
> > > >Solid, technical papers that explore new territory or point out new directions for research are preferable to papers that advance the state of the art, but only incrementally.
> > >
> > > [CVPR](https://cvpr2023.thecvf.com/Conferences/2023/ReviewerGuidelines#:~:text=not%20grounds%20for%20rejection%20by%20itself.) says:
> > > >A proposed method does not exceed the state-of-the-art accuracy on an existing benchmark dataset is not grounds for rejection by itself.
> > >
> > > [ACL](https://2023.aclweb.org/blog/review-acl23/#:~:text=SOTA%20results%20are%20neither%20necessary%20nor%20sufficient%20for%20a%20scientific%20contribution.) says:
> > > >SOTA results are neither necessary nor sufficient for a scientific contribution.
> > >
> > > In summary, **we are confident that our proposed method will be widely adopted instead of cosine distance** in few-shot learning, given its superior simplicity and effectiveness.

---

> > > > ### Comment · Reviewer_oKqH · 2023-08-21
> > > >
> > > > I thank the authors for their responses. I carefully read the response and find some of my concerns are addressed. However, my concern related to the motivation is alleviated but not fully addressed. Moreover, it is precisely because the authors only used an existing metric, Kendall's Rank Correlation, to replace cosine distance without providing a clearer explanation and analysis, the novelty and significance of the proposed method is limited. Thus, I do not think the paper is ready for publication and I consider not to change my rating.

---

> > > > > ### Author Response · Authors · 2023-08-21
> > > > > **Further Clarifications**
> > > > >
> > > > > Thank you for your feedback. We are pleased to note that some of your concerns have been addressed in our response. Regarding the concerns you mentioned in your response, we aim to provide you with a more comprehensive and systematic elucidation of the principles underpinning our method.
> > > > >
> > > > > We would like to first reiterate the background of the few-shot learning task. In this scenario, models are trained on base classes, and when faced with a novel few-shot task, the data belongs to entirely new classes that the model has never encountered during training. This means **there is no overlap between the data used for model training and the data used for testing.**
> > > > >
> > > > > This situation leads to a fundamental distinction between features of novel classes and base classes. As we mentioned earlier, **a crucial characteristic is that the features’ channels of novel class data possess smaller and closely clustered values.** This phenomenon is evident from the experimental results presented in Table C of the attached PDF, spanning various datasets with diverse domain disparities. It's observable that base class features possess noticeably larger variance, while new class features exhibit significantly smaller variance, indicating more concentrated and closer channel value distribution. This observation substantiates the universally valid nature of our finding. **We emphasize that this phenomenon has been novelly revealed in our research, whereas it remained unexplored in prior studies.** Such a phenomenon **leads to significant challenges when utilizing distance metrics like geometric similarity to calculate semantic similarity between features**, as exemplified earlier. Channel importance ranking, however, offers an effective solution to address this issue. Thus, we have considered **incorporating channel importance ranking into few-shot learning. Kendall’s rank correlation emerges as a suitable metric to gauge the consistency of rankings between any two channels, which aligns with our intuitive rationale for employing it in the realm of few-shot learning.**
> > > > >
> > > > > From the results in Table 1, it's evident that **employing Kendall’s rank correlation solely during the inference phase has already led to substantial improvements.** This improvement even surpasses the recent state-of-the-art method CIM, designed explicitly for inference in few-shot learning, undeniably showcasing the efficacy of Kendall’s rank correlation. However, due to the inherently non-differentiable nature of ranking computations, directly using Kendall rank correlation in meta-training is unfeasible. Furthermore, if employed solely during inference, the improvement achieved is constrained by the inconsistency between training and testing objectives.
> > > > >
> > > > > To address this issue, we want to emphasize that we have devised **differentiable Kendall’s rank correlation, a significant aspect of our approach.** We introduce the differentiable form of Kendall rank correlation and theoretically prove that this differentiable representation serves as a smooth approximation to the original Kendall's rank correlation. This innovation enables the integration of Kendall's rank correlation into the meta-training stage of few-shot learning. Extensive experimental results also confirm that in comparison to solely employing Kendall rank correlation during inference, the utilization of differentiable Kendall's rank correlation unquestionably results in further performance improvements. It also demonstrates that **our method can be effectively integrated with existing approaches, yielding significant improvements without introducing additional training costs.**
> > > > >
> > > > > We sincerely hope that this response addresses your concerns, and we are open to further discussion.
> > > > >
> > > > > ---
> > > > >
> > > > > In addition, given that our paper introduces a simple and effective algorithm based on empirical findings, presents comprehensive experiments, provides accessible code ensuring reproducibility, and raises no ethical concerns, **we believe it does not merit a reject status**, whose definition is
> > > > > >3: Reject: For instance, a paper with technical flaws, weak evaluation, inadequate reproducibility and incompletely addressed ethical considerations.

---

> > > ### Author Response · Authors · 2023-08-19
> > > **Further Clarifications (2/3)**
> > >
> > > *Q2: The authors didn’t explain clearly why does the Kendall's rank correlation have a decrease in performance for deeper backbones? In the response, the authors claimed expecting any model to achieve performance improvement without training by simply replacing cosine similarity at test time, would be impractical. I think there is a problem with the expression of this sentence. In fact, if the proposed similarity measure is effective, there will be a performance improvement for various backbones. This has nothing to do with the depth of the backbone.*
> > >
> > > Thank you for your insights. While we value your perspective, **we hold a different view on this matter: The effectiveness of a method can not translate to its consistent outperformance in all situations.**
> > >
> > > We'd like to emphasize that Table 1 is designed as an exploratory experiment, representing solely the utilization of Kendall rank correlation during the inference phase, without any incorporation of meta-training. Since the objectives of model **training and testing are not aligned**, there could be instances where employing simple Kendall rank correlation during the testing phase may not yield improved results. However, **when utilizing differentiable Kendall rank correlation for meta-training, a consistent performance enhancement can be achieved.** This observation becomes more evident from Figure 4, which illustrates the channel-wise ablation experiments. Here, the inclusion of differentiable Kendall rank correlation during training leads to a noteworthy enhancement in performance.
> > >
> > > As a similar scenario for reference, consider the use of **Euclidean distance** versus **cosine distance** in few-shot learning. While Euclidean distance was initially employed as the distance metric in ProtoNets, more recent methods have gravitated towards cosine distance. **Does this suggest that cosine distance invariably outperforms Euclidean distance in every few-shot learning scenario? Not necessarily.** But its wide acceptance doesn't diminish its relevance, especially if it demonstrates superior performance in a majority of cases or on average.
> > >
> > > Similarly, in our case, Table 1 clearly reveals that **the adoption of Kendall's rank correlation during the inference phase, in contrast to cosine similarity, yields significant improvements across multiple datasets with varying domains.** This improvement even extends to the latest specialized few-shot learning method for the inference phase, CIM. Even in the case of the deeper backbone networks you mentioned, Kendall's rank correlation actually exhibits noteworthy enhancement across five datasets with more substantial domain gaps.
> > >
> > > In summary, performance improvements across various scenarios are indeed observed. While there may be certain instances where outperformance is not evident, this doesn't diminish the effectiveness of the method. In fact, **we are confident that our approach will become a favored alternative to cosine distance in the future**.

---

> > > ### Author Response · Authors · 2023-08-19
> > > **Further Clarifications (3/3)**
> > >
> > > *Q3: I think the author still hasn't explained the motivation of the paper well, and more of it is the phenomenon observed in the experiment, without clarifying the reason behind it. If the authors can theoretically explain why Kendall's Rank Correlation can be used to replace cosine distance, it will greatly improve the quality of the paper. I think this is the core innovation of the paper.*
> > >
> > > First of all, we would like to emphasize that **our paper focuses on presenting an algorithm inspired by empirical observations**. While we acknowledge the importance of theoretical aspects you mentioned, they fall beyond the primary scope of our current work. Our key contribution is **pinpointing a novel phenomenon and subsequently designing a simple and effective algorithm based on this discovery**. We also demonstrate the efficacy of our approach and its ability to integrate effortlessly with various cosine-based methods without incurring extra training expenses. We view the theoretical exploration as an avenue for future research.
> > >
> > > Nevertheless, instead of a theoretical explanation, **our paper offers an intuitive understanding.** The motivation behind our proposed method stems from the observation that, in few-shot learning, models are not exposed to novel classes during training and naturally exhibit distinct feature extraction in comparison to the base classes used for model pre-training. Our approach reveals a novel observation: for a novel class, feature channels exhibit smaller and tightly grouped values compared to base classes, which we have substantiated as a universally valid conclusion. This situation creates a challenge in employing geometric similarity to accurately distinguish the importance among feature channels. The utilization of channel importance ranking, instead, offers an effective solution to this challenge. **We would like to emphasize that all of these aspects remain unexplored in prior research.**
> > >
> > > Furthermore, while we understand that providing a theory would enrich a paper, **its absence doesn't devalue the core innovation of our work**. For your reference, we would like to mention a highly renowned paper titled "The Lottery Ticket Hypothesis: Finding Sparse, Trainable Neural Networks." This paper observed the existence of "winning tickets" within deep neural networks, and its validation was conducted through the design of a search algorithm for these winning tickets. **Noteworthily, this paper abstained from delving deep into theoretical justifications.** Yet, would one argue that this omission detracts from its impact? Quite the opposite; the paper's straightforward and thought-provoking approach led to it being distinguished as the **"best paper" at ICLR 2019, attracting almost 2800 citations**.
> > >
> > > Additionally, we'd like to gently draw the reviewer's attention to the official [reviewer guidelines of NeurIPS](https://nips.cc/Conferences/2020/PaperInformation/ReviewerGuidelines#:~:text=may%20be%20theoretical), which state:
> > > >**There are many examples of contributions that warrant publication at NeurIPS.** These contributions **may be** theoretical, methodological, algorithmic, empirical, connecting ideas in disparate fields (“bridge papers”), or providing a critical analysis (e.g., principled justifications of why the community is going after the wrong outcome or using the wrong types of approaches.).
> > >
> > > In summary, while we value your feedback, we believe the absence of a theoretical explanation does not diminish the innovation and significance of this paper.
> > >
> > > ---
> > >
> > > We hope this response addresses your concerns, and **we are open to further discussion** to ensure the quality and clarity of our work.

---

> ### Author Response · Authors · 2023-08-15
> **We would be grateful if you could take a look at the response**
>
> Dear Reviewer oKqH:
>
> We sincerely appreciate your valuable time devoted to reviewing our manuscript. We would like to gently remind you of the **approaching deadline for the discussion phase**. We have diligently addressed the issues you raised in your feedback, providing detailed explanations. For instance, we have included comparative experiments with SOTA methods, demonstrating the enhanced performance when our approach is integrated with a stronger baseline, DeepEMD, through a straightforward substitution of cosine similarity with Kendall’s rank correlation. We have also addressed your confusion about the motivation behind our proposed method through the utilization of carefully considered language, along with more intuitive examples. Would you kindly take a moment to look at it?
>
> We are very enthusiastic about engaging in more in-depth discussions with you.

---

### Official Review · Reviewer_5zcq · 2023-07-06

**Soundness:** 4 excellent
**Presentation:** 3 good
**Contribution:** 3 good
**Rating:** 6
**Confidence:** 4

**Summary:**

The authors suggest a new similarity metric that utilizes differentiable Kendall’s rank correlation instead of the commonly used geometric similarity metrics like cosine similarity in few-shot learning. By levitating the importance of small-valued feature channels, the proposed approach significantly improves the few-shot performance across multiple datasets from various domains.

**Strengths:**

- The presented idea is simple and appears to be effective, and the overall approach is clearly expressed. The work is built on a simple yet powerful observation about the feature activation statistics (and their differences on base vs novel classes).
- Utilizing a differentiable version of Kendall’s rank coefficient measure as an alternative similarity metric is an original idea to the best of my knowledge.
- The proposed method enables efficient end-to-end few-shot learning without introducting problematic hyper-parameters.
- It is impressive that using the proposed metric directly for testing, even without any pre-training, improves the result.
- The proposed approach has proven to be effective in many common few-shot datasets across various domains, outperforming competitive methods in terms of performance improvements.
- Ablation studies provide valuable insights that demonstrate the effectiveness of the proposed approach.

**Weaknesses:**

- While a solution to the Figure 1 observation is proposed based on Kendall’s rank correlation, I am not sure if this is the most simple way to handle the problem. In particular, could a simple instance-statistics-driven normalization scheme, such as group norm or instance norm or similar, could address the feature-scale issues?
- Similarly, can’t the cosine similarity, combined with temperature scaling (a well known practice in , see Baseline++ paper for a detailed discussion) , where temperature may be different at test time, also address the problem of scale?
- The explanations on Kendall’s rank correlation can be extended to make it more explanatory and to shed even more light on its complexities to make the paper more self-contained.

**Questions:**

- The paper does a great job in pointing out a source of problem in few-shot classification and a good job in proposing a way to address it. However, it feels somewhat weak in terms of looking in-depth into the problem. Following the ‘weaknesses’ discussion above, it would have been great to improve the paper on this end and explore the advantages/disadvantages of some simple potential alternative fixes such as (i) instance/group normalization, (ii) temperature scaling combined with l2 normalization, or (iii) a simple attention mechanism such squeeze&excitation attention.
- Suggestion: The name of the dataset on which the presented results are obtained can be added in the discussions (or captions) of Figures 5 and 6.
- Suggestion: The overall figure quality can be improved.

**Limitations:**

No additional comments.

---

> ### Author Rebuttal · Authors · 2023-08-09
>
> Q1: Could a simple instance-statistics-driven normalization scheme, such as group norm or instance norm or similar, could address the feature-scale issues?
>
> A1: **No.** Actually, solely adopting Kendall's rank correlation during the inference stage far exceeds what can be achieved by employing simple feature scaling methods, and we conduct experiments to demonstrate this. Specifically, on top of the raw features from the model's output, we apply GroupNorm, InstanceNorm, and squeeze&excitation attention for feature scaling transformations, followed by classification using cosine similarity (Please find detailed results in Table B of the attached PDF in our "global" response).
>
> Regarding **GroupNorm**, we test two values for the "num_group" parameter, namely 16 and 32. Although its performance occasionally surpasses the original cosine similarity, we observe that it is generally inferior to Kendall's rank correlation in terms of performance. Concerning **InstanceNorm**, we notice that this operation results in a substantial performance decline, with a decrease of over 10% compared to directly using cosine similarity. Furthermore, we also explore the use of **Squeeze&Excitation Attention**. We integrate this module into the backbone network for training, and the results also show that it does not demonstrate a clear advantage.
> Actually, in Table 1 of Section 4, we have compared our method with the **CIM** method, which is a most recently proposed simple test-time feature scaling method in few-shot learning. Across multiple datasets with diverse domain differences, our method also consistently outperforms CIM. These findings provide strong evidence that the improvements obtained through Kendall ranking correlation are not attainable merely by simple feature scaling. We will include this discussion in the paper to emphasize the superiority of our proposed method. Thank you for your valuable review.
>
> Q2: Can’t the cosine similarity, combined with temperature scaling also address the problem of scale?
>
> A2: **No, it can't.** In fact, the temperature coefficient is also used in the meta-baseline method to adjust the output probabilities. When reproducing their experimental results, we follow the original settings of the meta-baseline and set this parameter as learnable during training. We conduct experiments accordingly, and if we solely adjust this parameter, the performance would be lower compared to the results reported in the meta-baseline paper, as shown below.
>
> | Method   | Backbone |   T: Learnable | T= 1 |T= 0.1 |T= 0.01 |
> | :---: | :---: | :---: |:---: |:---: |:---: |
> |Meta-Baseline (cosine)|ResNet-12|63.17|62.35|62.84|62.76|
>
> Clearly, adjusting the temperature parameter alone cannot achieve the level of performance improvement obtained by using Kendall’s ranking correlation.
>
> Q3: The explanations on Kendall’s rank correlation can be extended to make it more explanatory and to shed even more light on its complexities to make the paper more self-contained.
>
> A3: Thank you for bringing this to our attention. Let us provide a detailed explanation below.
>
> Our approach emerged from observing an apparent difference in feature channel values between base data and novel data. We found that compared to base classes, when the feature extractor faces a novel class that is unseen before, the feature channel values become more uniform, i.e., **for a novel class, most non-core features' channels have small and closely clustered values** in the range [0.25, 0.5] (see Figure 1 of the submitted paper).
>
> This phenomenon occurs because the model is trained on the base data, and consequently exhibits reduced variation of feature values when dealing with novel data. This situation creates a challenge in employing geometric similarity to accurately distinguish the importance among non-core feature channels.
>
> To provide a concrete example, consider distinguishing between dogs and wolves. **While they share nearly identical core visual features, minor features play a vital role in differentiating them**. Suppose the core feature, two minor features are represented by channels 1, 2, and 3, respectively, in the feature vector. A dog prototype may have feature (1, 0.3, 0.2), and a wolf prototype may have feature (1, 0.2, 0.3). Now, for a test image with feature (0.9, 0.28, 0.22), it appears more dog-like, as the 2nd feature is more prominent than the 3rd. However, cosine distance misleadingly places this test image closer to the wolf prototype (distance=0.031) rather than the dog prototype (distance=0.048).
>
> Contrastingly, the test image shares the same channel ranking (1, 2, 3) as the dog prototype, whereas the wolf prototype's channel ranking is (1, 3, 2). Inspired by this, we employ Kendall’s rank correlation to more accurately discern between dogs and wolves, highlighting the utility of our approach.
>
> We hope this clarification better conveys the underlying rationale for our method, and we will carefully review this section in the revised paper.
>
> Q4: It would have been great to explore the advantages/disadvantages of some simple potential alternative fixes such as (i) instance/group normalization, (ii) temperature scaling combined with l2 normalization, or (iii) a simple attention mechanism such squeeze&excitation attention.
>
> A4: Thank you for the comment. In the response to Q1 and Q2, we have verified and demonstrated that the performance improvement achieved by using Kendall's rank correlation far exceeds what can be achieved by employing simple feature scaling methods, which is related to this question.
>
> Suggestion: i) The name of the dataset on which the presented results are obtained can be added in the discussions (or captions) of Figures 5 and 6. ii) The overall figure quality can be improved.
>
> A5: Thank you very much for your valuable feedback. We will make the necessary revisions to the manuscript according to your suggestions.

---

> > ### Comment · Reviewer_5zcq · 2023-08-16
> >
> > I am quite happy with the rebuttal’s responses and I value this paper not only for its novel technique, but also (or perhaps firstly) for its scientific contribution towards demystifying the few-shot learning’s challenges. I have increased my score to weak accept. I have also looked at the lower-rating reviews, while I appreciate them, I haven’t seen any criticism that leads to changing my mind; however they do have good points & suggestions.

---

> > > ### Author Response · Authors · 2023-08-17
> > >
> > > Thank you very much for taking the time to re-evaluate our paper after considering our rebuttal. We greatly appreciate your positive remarks regarding the novel technique we introduced and our contribution to shedding light on the challenges of few-shot learning. We are also grateful for the improved score you've given our work. Your constructive feedback and acknowledgement of our efforts is truly encouraging. We assure you that the valuable suggestions and insights from you and other reviewers will certainly be integrated into our revised version.

---

### Official Review · Reviewer_7vc7 · 2023-07-12

**Soundness:** 3 good
**Presentation:** 3 good
**Contribution:** 2 fair
**Rating:** 4
**Confidence:** 5

**Summary:**

This paper aims at addressing the uniform distribution of values across features on novel classes, and proposes to use Kendall's rank corerlation instead of geometric similarity metrics to improve the performance of few-shot learning pipelines. Moreover, the authors propose a designed loss for meta-training to make Kendall's rank correlation differentiable. The experimental results demonstrate the usefulness of the propose method.

**Strengths:**

- The idea is technically sound. Theoretically, the channel importance is definitely beneficial to improve the accuracy of FSL.
- The expeirment results show the usefulness of the propose method.
- The presentation is well and easy to follow.

**Weaknesses:**

- The novelty and significance of the proposed method is roughly limited. The core idea of Kendall's rank correlation is to measure the consistency of pairwise rankings for each channel pair. However, such a issue has been considered in previous cross-matching related works such as DeepEMD. If this design can be incorporated with DeepEMD or other cross-matching pipelines, it would be interesting to see the overall performance and the conclusions will be more convincing.
- The experimental results is not impressive enough. Though this method achieves improvements compared to metric-based methods, its performance is still below the SOTA.
- In figure 1, it is proper to show more cases to demonstrate the advanceness of Kendall's rank correlation.

**Questions:**

- It seems the work is based on prototype. Is it possible to perform it between support-query pairs rather than support proto-query pairs? As it highlights the importance of feature channels, it should be more meaningful to perform support-query correlation.

**Limitations:**

Please refer to the weaknesses.

---

> ### Author Rebuttal · Authors · 2023-08-09
>
> Q1: The proposed method's novelty is limited, since the consistency of pairwise rankings for each channel pair has been considered in previous works such as DeepEMD. Can this design be incorporated with DeepEMD?
>
> A1: Thanks for your insightful feedback. While it may seem that both DeepEMD and our method consider the consistency of pairwise rankings, they do so from **two different perspectives**. Specifically, DeepEMD employs the Earth Mover's Distance to match various **local regions** in the spatial domain of the image. In contrast, our method leverages Kendall’s Rank Correlation to achieve consistency across **feature channels**.
>
> Moreover, we would like to highlight that our method can easily be integrated into existing methods **without increasing training costs**. As an example, **our method can indeed be incorporated with DeepEMD** by substituting the cosine similarities used therein with Kendall’s Rank Correlations. Through this modification, we have observed **a large improvement for DeepEMD** (1%-2%). All the latest experimental results, demonstrating this enhancement, are detailed in Table A of the attached PDF in our "global" response.
>
> Q2: The experimental results are not impressive enough. Though this method achieves improvements compared to metric-based methods, its performance is still below the SOTA.
>
> A2: Thanks for bringing this to our attention. It's important to clarify that our current experiments are conducted **based on a simple and widely-adopted baseline** (meta-baseline) to **demonstrate the effectiveness** of our method. While this may lead to performance that is below the SOTA, we believe it adequately demonstrates the capabilities and potential of our method.
>
> Moreover, as mentioned in the response to the previous question, our method could be easily integrated with existing methods. **Combining our method with a stronger baseline**, DeepEMD, we can **achieve the current SOTA performance**, as shown in Table A of the attached PDF in our "global" response.
>
> Furthermore, even for most recently proposed methods such as CIM, we have shown that simply replacing cosine similarity with Kendall’s Rank correlation (our method) at the inference stage can result in significant improvements across various datasets with domain differences (see Table 1 of the submitted paper). This indicates that **our method is ready for integrating with future SOTA methods** to achieve additional improvement.
>
> We hope this explanation can address your concern, and provide a clearer demonstration of the value and adaptability of our method.
>
> Q3: In figure 1, it is proper to show more cases to demonstrate the advanceness of Kendall's rank correlation.
>
> A3: Thank you for this feedback. Indeed, there are numerous such examples. We would like to kindly remind you that, in the **supplementary material's visual analysis section**, we provide more intuitive illustrations of the superior performance of Kendall’s ranking correlation.
>
> Concretely, we employ Kendall’s Rank Correlation and cosine similarity to visualize the feature maps of the query samples with the aim of confirming the accurate localization of salient objects in the images. It is evident that the utilization of Kendall’s Rank Correlation results in a more precise localization of the distinctive regions within the query sample.
>
> Moreover,  we also conduct an in-depth visual analysis involving channel ablation. It is noticeable that the discriminative key features of the dog predominantly exist in channels with lower values. Utilizing Kendall's rank correlation effectively captures these essential features, whereas cosine similarity disregards them, providing evidence of the effectiveness of our method.
>
> Q4: It seems the work is based on prototype. Is it possible to perform it between support-query pairs rather than support proto-query pairs? As it highlights the importance of feature channels, it should be more meaningful to perform support-query correlation.
>
> A4: Thanks for your insightful feedback. Indeed, our method can be applied to not only prototype-query matching **but also support-query matching**. For example, we replace cosine similarity with Kendall’s Rank Correlation in InfoPatch which is a **contrastive-learning-based few-shot learning method**, and observe an improvement in both 1-shot and 5-shot tasks on mini-ImageNet, as shown in Table A of the attached PDF in our "global" response.

---

> ### Author Response · Authors · 2023-08-15
> **We would be grateful if you could take a look at the response**
>
> Dear Reviewer 7vc7:
>
> We sincerely appreciate your valuable time devoted to reviewing our manuscript. We would like to gently remind you of the **approaching deadline for the discussion phase**. We have diligently addressed the issues you raised in your feedback, providing detailed explanations. For instance, we have elucidated that our approach and DeepEMD, in fact, address the challenges of few-shot learning from two distinct perspectives. Moreover, we have conducted experiments that demonstrate the integration of our method with DeepEMD. By straightforwardly substituting the cosine similarity in DeepEMD with Kendall’s rank correlation, we have successfully achieved state-of-the-art performance. Would you kindly take a moment to look at it?
>
> We are very enthusiastic about engaging in more in-depth discussions with you.

---

### Author Rebuttal · Authors · 2023-08-09

We sincerely appreciate the valuable comments from all reviewers. We have endeavored to provide explanations for the questions raised in the respective comments section.

Additionally, the supplementary experiments have been incorporated into the PDF attached to the global response. Specifically, the appended PDF encompasses the following:
1. **Comparison with state-of-the-art methods** when integrating our approach with DeepEMD and InfoPatch. This provides evidence that when integrating our method with stronger baselines, it is capable of achieving state-of-the-art (SOTA) performance.
2. **Comparison among Kendall's rank correlation, distances other than cosine similarity, and simple feature scaling transformation modules.** This highlights the superiority of Kendall's rank correlation over alternative distances in few-shot learning, while also confirming that this improvement cannot be surpassed by mere simple feature scaling methods.
3. **Comparison of the variance of feature channel values between the base dataset and various novel datasets.** This validates the widely applicable conclusion that features' channel values exhibit a high degree of clustering and similarity for novel classes unseen by the model.
4. **Additional experiments of Kendall's rank correlation on 5-way 5-shot tasks** across multiple datasets with diverse domain variations.

---

### Author Response · Authors · 2023-08-14
**Seeking Your Further Feedback**

Dear AC and Reviewers,

We would like to express our sincere gratitude for the time and effort you have dedicated to the evaluation of our work. Your valuable insights and expertise are deeply appreciated.

In response to your review comments, we have prepared detailed answers to address the concerns and inquiries you have regarding our paper. **Should there be any unresolved issues, or should you need further clarification or have additional questions, please do not hesitate to let us know**. We stand ready to provide any information or clarification that may assist you in your review.

Thank you once again for your invaluable time and consideration. We eagerly look forward to hearing from you, and we hope to have the opportunity to respond to any feedback or questions you may have.

With warm regards,

Authors

---

### Decision · Program_Chairs · 2023-09-21

**Decision:**

Accept (poster)

**Comment:**

This paper obtained diverging reviews: 1 weak accept, 1 borderline accept, 2 borderline reject, and 1 reject. While two reviewers appreciated the simple yet effective idea and the consistent improvement over standard methods, the other reviewers raised concerns about limited novelty, unclear motivation, lack of theoretical justification, and missing comparisons/analyses. The authors provided very detailed rebuttals, addressing most of them with additional experiments/analyses. Despite the authors’ constant rebuttals and requests for discussion, some reviewers remain negative without responding to the authors. AC thus carefully examined the reviews, the rebuttal, and the discussion. Checking the details, AC finds that the raised concerns are mostly resolved. For example, the authors clarified their motivation based on the observation about feature channel distributions, explained the difference of the proposed Kendall's rank correlation from DeepEMD, and provided SOTA results when combined with DeepEMD. The lack of theoretical motivation may be a drawback, but the authors’ empirical justification is intuitive and reasonable enough. In particular, AC agrees with reviewer 5zcq that the differentiable version of Kendall’s rank correlation as an alternative similarity metric is an original idea and can be very useful for metric-based few-shot classification and beyond. Therefore, AC recommends acceptance.